# FIARSE: Model-Heterogeneous Federated Learning via Importance-Aware Submodel Extraction

**Feijie Wu[1], Xingchen Wang[1], Yaqing Wang[2], Tianci Liu[1], Lu Su[1], Jing Gao[1]**
[1]Purdue University    [2]Google DeepMind
{wu1977, wang2930, liu3351, lusu, jinggao}@purdue.edu
yaqingwang@google.com

## Abstract

In federated learning (FL), accommodating clients' varied computational capacities poses a challenge, often limiting the participation of those with constrained resources in global model training. To address this issue, the concept of model heterogeneity through submodel extraction has emerged, offering a tailored solution that aligns the model's complexity with each client's computational capacity. In this work, we propose Federated Importance-Aware Submodel Extraction (FIARSE), a novel approach that dynamically adjusts submodels based on the importance of model parameters, thereby overcoming the limitations of previous static and dynamic submodel extraction methods. Compared to existing works, the proposed method offers a theoretical foundation for the submodel extraction and eliminates the need for additional information beyond the model parameters themselves to determine parameter importance, significantly reducing the overhead on clients. Extensive experiments are conducted on various datasets to showcase the superior performance of the proposed FIARSE.

## 1 Introduction

Federated learning (FL) [36, 52] stands out as a promising distributed training paradigm, in which the clients enjoy mutual information without jeopardizing data privacy. Specifically, the FL server requests the clients to train a model with their local data and aggregates the models into a global one. Such a paradigm, however, may fail in a real-world FL system, where the clients usually have varying computation capacities [13, 42, 46], likely preventing the clients with insufficient computation resources from being involved in training a large global model [73].

To tackle the challenge, a practical solution is to enable model heterogeneity, ensuring that the model deployed on each individual client aligns with its local computation capacity. This can be done by extracting a submodel for each client from the global model, which encompasses a subset of the parameters of the global model. During the model training period, the parameters of each submodel are thereby retrieved from the counterpart of the global model. Importantly, each parameter of the global model is exclusively averaged among the submodels containing it [13].

Depending on whether the submodels undergo reconstruction during the model training process, existing works fall into two categories: static submodel extraction [13, 25, 33, 35, 58, 67, 76] and dynamic submodel extraction [2, 7, 8, 21, 45].

Static submodel extraction creates a submodel for each client prior to the model training process. As illustrated in Figure 1a, the submodel remains unchanged throughout the training process. However, static submodel extraction suffers from certain limitations that affect both local clients and global performance. Locally, the metrics or knowledge used to extract a submodel for each client are likely to evolve during the training process. Since static submodels do not account for this evolution, they may fail to achieve optimal performance. Globally, the extractions of submodels from certain parts may incur client drift during the training, as pointed out by Alam et al. [2] and Liao et al. [45]. This

---

38th Conference on Neural Information Processing Systems (NeurIPS 2024).

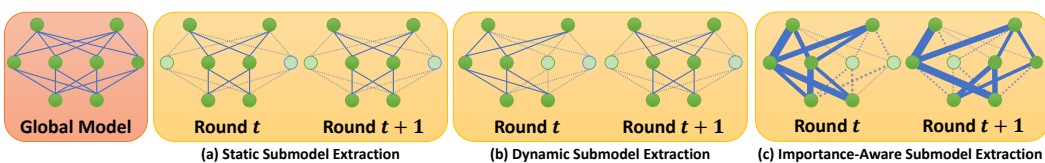

Figure 1: Three types of submodel extraction for model training, i.e., static, dynamic, and importance-aware (ours). The figure demonstrates the global model on the server and the local models of two consecutive rounds on a client. Note that solid lines represent the parameters preserved in the local model, while dash lines indicate the parameters excluded from the local model. In importance-aware submodel extraction, we present the importance of the parameters via the line thickness .

phenomenon emerges when the clients update their submodels biased to their local optima, thereby deviating from the global optimum. As a result, it either degrades the training efficiency or leads to a surrogate convergence on a global scale [34, 64].

Dynamic submodel extraction updates each submodel dynamically in each training round, and thus is able to capture the evolution of the global model. For example, FedRolex [2] employs a rolling-based submodel extraction approach, addressing client drift by ensuring equal chances for training each parameter as shown in Figure 1b. While the method demonstrates significant performance improvement over static submodel extraction in terms of the global model, it sacrifices the submodels' performance. This is because FedRolex treats every parameter equally, leading to a lack of clear guidance on submodel extraction.

To overcome the limitations of the above methods, in this paper, we propose Federated Importance-AwaRe Submodel Extraction, named FIARSE, for model-heterogeneous FL. Specifically, the proposed FIARSE method extracts the submodels based on the importance levels of model parameters. Here important parameters are the edges of the neural network that can induce dramatic changes in the final outputs when removed.

Figure 1c visually illustrates the submodels extracted by FIARSE. As demonstrated, FIARSE constructs a submodel by sequentially incorporating parameters in descending order of their importance levels (represented by the thickness of the edges in the model), from highest to lowest, until the client's maximum computation capacity is reached. In contrast to static submodel extraction, our approach enables dynamic updates of the submodels, thereby effectively capturing the evolving nature of model parameters. When compared to rolling-based submodel extraction, FIARSE adaptively identifies important parameters, ensuring outstanding performance for both the global model and local submodels. Referring back to the model shown in Figure 1, the edges connecting the leftmost neuron at the second layer are indicative of important parameters. However, the rolling-based submodel extraction method would roll these parameters out of the submodel in round $t + 1$ (as shown in Figure 1b), resulting in inadequate training on these crucial parameters. In contrast, our proposed method can identify and retain these important parameters in the training process, as illustrated in Figure 1c.

**Contributions.**  The contributions of this paper can be highlighted from the following perspectives:

- **Algorithmically**, we propose an importance-aware framework for model-heterogeneous federated learning. This framework can construct a client-specific submodel calibrated to each client's computation and storage capacity. It achieves this by representing the importance level of each model parameter with its magnitude, thereby avoiding additional storage or computational overhead needed to explicitly maintain the importance scores.

- **Theoretically**, we prove that the proposed algorithm converges at a rate of $O\left(1/\sqrt{T}\right)$, where $T$ is the number of communication rounds. This convergence rate is consistent with that of the state-of-the-art FL algorithms, indicating that the proposed submodel construction mechanism does not undermine the convergence properties. To the best of our knowledge, this is the first study to provide a theoretical analysis for model-heterogeneity FL under partial-client participation.

- **Empirically**, we conduct extensive experiments on image and text classification tasks, employing a training-from-scratched model ResNet, and a pretrained model RoBERTa. The results verify that FIARSE significantly outperforms existing approaches, particularly on the clients with limited capacity. The superior performance on resource-constrained devices demonstrates FIARSE's advantages in efficiently adapting submodels to meet diverse capabilities.

## 2 Related Work

This section discusses the state-of-the-art works that are most relevant to our research. Appendix A provides a more comprehensive review.

**Computation Heterogeneity in FL.** Computation heterogeneity in FL refers to the varying computational capacities among clients, including differences in hardware capabilities and resource availability. One typical solution is to allow faster clients to perform more local updates, while slower ones update their local models fewer times [43, 50, 53, 57, 64, 72]. However, these approaches require clients to train the full model, which becomes infeasible when some clients cannot load the full model due to limited computation resources. In our work, we extract a submodel for each client that fits within their computational capacity.

**Model Customization in FL.** Model customization allows the clients to build their local models that align with their local computation resources [11, 28, 46, 66, 77, 80]. To aggregate these heterogeneous models, the method often employs distillation for knowledge transfer, which requires a shared public dataset. However, this approach becomes infeasible when a shared public dataset is unavailable [3, 59]. In our work, we eliminate the need for a public dataset, broadening its applicability to a wider range of scenarios.

**Model Sparsification in FL.** Model sparsification, also known as model pruning, removes the unimportant model parameters from a deep learning model, reducing computation overhead and tailoring model sizes to suit clients with varying computational resources [8, 45, 81]. For example, Flado [45] achieves model sparsification by requiring each client to maintain the importance levels of model parameters and extracting a submodel that encompasses the most important model parameters. While effective, this method incurs considerable overhead on each client in terms of both storage and computation, posing significant challenges for resource-constrained devices. In contrast, our work implicitly represents parameter importance through their values, eliminating the need to maintain separate importance scores for each parameter.

## 3 Preliminary: Model-Heterogeneous Federated Learning

**Problem Formulation.** Consider there are $N$ clients in an FL system. The computation capacity of each client $i \in [N]$ is metered by the maximum ratio of a submodel extracted from the global model $\tilde{\boldsymbol{x}} \in \mathbb{R}^d$ and denoted by $\gamma_i \in [0, 1]$, and the values of $\gamma_i$ could vary among the clients. To enable submodel extraction, each client $i$ should assign a binary mask $\mathcal{M}^{(i)} \in \{0, 1\}^d$ such that $\|\mathcal{M}^{(i)}\|_1 \leq \gamma_i d$. Let $\mathcal{M}$ be the collections of the clients' masks $\mathcal{M}^{(i)}$, i.e., $\mathcal{M} = \cup_{i \in [N]} \mathcal{M}^{(i)} \in \{0, 1\}^{N \times d}$. To simultaneously optimize the model parameter and the clients' masks, the model-heterogeneity FL system is formulated as

$$\min_{\tilde{\boldsymbol{x}} \in \mathbb{R}^d, \mathcal{M} \in \{0,1\}^{N \times d}} F(\tilde{\boldsymbol{x}}, \mathcal{M}) \triangleq \frac{1}{N} \sum_{i \in [N]} \left[ F_i \left( \tilde{\boldsymbol{x}} \odot \mathcal{M}^{(i)} \right) \triangleq \mathbb{E}_{b \sim \mathcal{D}_i} \ell \left( \tilde{\boldsymbol{x}} \odot \mathcal{M}^{(i)}; b \right) \right]. \quad (1)$$

Let $\mathcal{D}_i$ be the local dataset of client $i \in [N]$, $\ell$ be the loss function which calculates the loss for a model on a given data sample (including an input and a target). Therefore, the local objective $F_i(\cdot)$ in Equation (1) indicates the expected loss for client $i$ on the local dataset. For simplicity, we consider all $N$ clients to carry equal weights (i.e., $1/N$) in Equation (1), and the proposed approach can be extended to the scenario where the clients are with different weights.

**A Generic Solution: Partial Averaging.** There are numerous submodel extraction approaches, which are categorized into static and dynamic submodel extractions. However, these methods adopt partial averaging to aggregate clients' models into a global one, and the details are outlined as follows: At round $t \in \{0, 1, \dots\}$,

- **Sampling:** The server randomly samples a subset of clients $\mathcal{A} \subset [N]$ and distributes the global model parameters $\tilde{\boldsymbol{x}}_t$ to the selected clients.

- **Local Model Training:** The clients $i \in \mathcal{A}$ performs $K$-times local updates via $\boldsymbol{x}_{t,k}^{(i)} = \boldsymbol{x}_{t,k-1}^{(i)} - \eta \nabla F_i \left( \boldsymbol{x}_{t,k-1}^{(i)} \odot \mathcal{M}_t^{(i)} \right) \odot \mathcal{M}_t^{(i)}$, where $k \in \{1, \dots, K\}$ and $\boldsymbol{x}_{t,0}^{(i)} = \tilde{\boldsymbol{x}}_t$. It is noted that $\mathcal{M}_t^{(i)}$ represents a binary mask for client $i$ at $t$-th round, which can be either predefined [2, 13, 25, 33] or determined by the client [21, 45, 76].

- **Global Model Aggregation:** The clients collect the model updates from the participants $\mathcal{A}$ and perform the global model aggregation via $\tilde{\boldsymbol{x}}_{t+1} = \tilde{\boldsymbol{x}}_t - \eta_s \mathsf{Agg}_{i \in \mathcal{A}} \left( \boldsymbol{x}_{t,K}^{(i)} - \tilde{\boldsymbol{x}}_t \right)$. Given a set of $d$-dimension vectors $\boldsymbol{v}^0, \ldots, \boldsymbol{v}^{|\mathcal{A}|-1} \in \mathbb{R}^d$, $\mathsf{Agg}_{i \in \mathcal{A}}(\boldsymbol{v})$ is defined as: (i) For the $j$-th index, $\mathsf{Agg}_{i \in \mathcal{A}}(\boldsymbol{v}_j) = \left( \sum_{i \in \mathcal{A}} \boldsymbol{v}_j^i \right) / \left( \sum_{i \in \mathcal{A}} \mathbf{1}\{\boldsymbol{v}_j^i \neq 0\} \right)$; (ii) $\mathsf{Agg}_{i \in \mathcal{A}}(\boldsymbol{v}) = \cup_{j \in [d]} \mathsf{Agg}_{i \in \mathcal{A}}(\boldsymbol{v}_j)$.

**Limitations.** The above solution adopts a consistent mask during local model training, which cannot obtain the optimal mask $\mathcal{M}$ for various clients as Equation (1) expects. Some recent works (e.g., Flado [45] and pFedGate [8]) have proposed to capture the importance levels of each model parameter in achieving the objective of Equation (1), where a submodel consists of the most important parameters up to the maximum capacity of a client. In these works, the clients hold the importance scores for each model parameter and extract a submodel accordingly. After the local training of the submodel, the clients take an additional step to optimize the importance scores. Despite the effectiveness of these approaches, their feasibility is compromised due to the massive costs related to storage and computation. These approaches entail a minimum training memory and storage of $O(d)$ and $O(d)$ on client $i \in [N]$, respectively, while HeteroFL [13] ensures the training memory within $O(\gamma_i d)$ and does not require additional storage. Moreover, it is time-consuming to separate model parameters update and mask optimization into two steps. A work [81] attempts to simplify the process by means of greedy pruning, where a submodel consists of the parameters selected from the largest to the smallest absolute values. Apparently, this approach avoids mask optimization while evolving the submodel architectures since they are associated with model parameters. However, this work keeps the mask consistent during local model training, which does not make sense because an update of model parameters should lead to a different mask.

## 4    FIARSE

**Solution Overview.** In this work, we explore the correlation between the value of model parameters and their importance levels, leveraging the insights from previous research [29, 54]. These studies reveal that the magnitude of model parameters can act as an indicator of their importance levels. This discovery offers an opportunity to simplify Equation (1). Given that our objective is to extract important parameters for submodel construction, we can approximate this goal by selecting larger parameters to build the submodel, rather than explicitly maintaining an importance score for each parameter. Though simplified, the problem is still challenging since the model parameters themselves are also variables to be optimized. To address this challenge, in Section 4.1, we will present a novel submodel construction method that can jointly select and optimize model parameters.

Finally, Section 4.2 introduces our proposed FL algorithm FIARSE that seamlessly integrates the submodel construction method and optimizes the global model. In detail, the clients optimize the model parameters $\tilde{\boldsymbol{x}}$ by leveraging the submodel construction method. The server subsequently aggregates these optimized models from the clients and initiates a new training round. Given that the collected model parameters inherently reflect their importance levels, the aggregated global model parameters also effectively capture their importance from a global perspective. Algorithm 1 concisely presents the pseudocode of FIARSE.

### 4.1    Submodel Construction

As highlighted in the overview, the insight that the values of model parameters are correlated with their importance allows us to reframe the problem of submodel construction. Intuitively, by controlling the number of parameters included in the submodel, we can ensure the computation and/or storage costs of the submodel not exceed the budget of the clients. To achieve this, we establish a threshold for the model parameters based on each client's capacity. Only those parameters whose values exceed this threshold are included in the respective client's submodel. Thanks to the correlation between a parameter's value and its importance level, this approach ensures that the parameters included in the submodel are of greater importance than those that are excluded. This idea can be implemented by converting the mask variable in Equation (1) into a function of the parameter values:

$$\min_{\tilde{\boldsymbol{x}} \in \mathbb{R}^d} F(\tilde{\boldsymbol{x}}, \mathcal{M}(\tilde{\boldsymbol{x}})) \triangleq \frac{1}{N} \sum_{i \in [N]} F_i \left( \tilde{\boldsymbol{x}} \odot \mathcal{M}^{(i)}(\tilde{\boldsymbol{x}}) \right), \quad \text{where} \quad \mathcal{M}^{(i)}(\tilde{\boldsymbol{x}}) = \begin{cases} 1, & |\tilde{\boldsymbol{x}}| \geq \theta_i \\ 0, & |\tilde{\boldsymbol{x}}| < \theta_i \end{cases}, \quad (2)$$

where $\mathcal{M}^{(i)}(\cdot)$ represents the mask function of client $i \in [N]$ and incorporates the threshold $\theta_i$ on a given model such that $\| \mathcal{M}^{(i)}(\tilde{\boldsymbol{x}}) \|_1 \leq \gamma_i d$; and $\mathcal{M}(\cdot)$ is the collections of all local mask

functions. As seen, the problem projects parameter importance to parameter values and thus can achieve parameter selection and model training through optimizing solely the parameter values.

Now the question is how to determine the threshold for Equation (2). In general, the threshold abides by the clients' local computation/storage capacity. Towards this end, we determine the value using $\mathsf{TopK}_\gamma(\cdot)$ operation, which selects the top $\gamma$ values of the given vector. For simplicity, we discuss model-wise threshold selection strategies in this section, where the threshold $\theta_i$ is set for $\mathsf{TopK}_{\gamma_i}(|\tilde{\boldsymbol{x}}|)$. Our proposed method is applicable for settings where different thresholds are assigned to different model parameters. Additional threshold selection strategies will be explored in Appendix D.1.

**Threshold-Controlled Biased Gradient Descent (TCB-GD).** We enhance Equation (2) by integrating straight-through estimation (STE) [4, 49], where we assume the mask is labeled with 1 with a probability determined by $\mathsf{clip}\left(\frac{|\tilde{\boldsymbol{x}}_j| - \theta_i}{|\tilde{\boldsymbol{x}}_j| + \theta_i}, 0, 1\right)$, where $\tilde{\boldsymbol{x}}_j$ means $j$-th element of a $d$-dimension model parameter $\tilde{\boldsymbol{x}}$. Therefore, the gradient calculated in the backward propagation on client $i \in [N]$ process adheres to:

$$\nabla_{\tilde{\boldsymbol{x}}} F_i\left(\tilde{\boldsymbol{x}} \odot \mathcal{M}^{(i)}(\tilde{\boldsymbol{x}})\right) = \underbrace{\nabla F_i\left(\tilde{\boldsymbol{x}} \odot \mathcal{M}^{(i)}(\tilde{\boldsymbol{x}})\right) \odot \mathcal{M}^{(i)}(\tilde{\boldsymbol{x}})}_{\text{Threshold-controlled}} \odot \underbrace{\left(1 + \frac{2|\tilde{\boldsymbol{x}}|\theta_i}{(|\tilde{\boldsymbol{x}}| + \theta_i)^2}\right)}_{\text{Biased}}, \quad (3)$$

A detailed derivation of the above equality is provided in Appendix B. There are two key differences in comparison with the gradient computation used by local training of partial averaging, i.e., $\nabla F_i\left(\tilde{\boldsymbol{x}} \odot \mathcal{M}^{(i)}\right) \odot \mathcal{M}^{(i)}$. First, the mask shifts with the model parameters changing. Second, the backward propagation considers the importance levels of model parameters and forms a biased gradient descent. This means the second term tries to make a clear border between the important and non-important parameters.

**Effectiveness.** We analyze our proposed approach based on its two features, namely, threshold-controlled and biased gradient computations:

• **Threshold-controlled:** By comparing Equation (3) with Equation (2), we notice that the parameters no less than the designated threshold will be updated. In other words, this gradient descent method only updates the parameters that are greater or equal to the given threshold, and those parameters that are initially smaller than the threshold never get updated. Obviously, the computation cost at each iteration remains constant or even smaller than the cost at the previous iterations.

  FL clients usually update the model for multiple iterations. According to the description above, the trained submodel is shrinking because some parameters may drop behind the threshold, while no new parameters are introduced to the submodel. Therefore, the proposed gradient descent method keeps the computation cost constant or even smaller than our expectation.

• **Biased:** Biasedness accelerates the update of the parameters near the threshold to distinguish their importance. In other words, less important parameters drop below the threshold and roll out, while the important ones continue to increase until stable. This feature guarantees the model parameters reflect their importance value by minimizing the existence of ambiguous parameters close to the threshold. In other words, the clients can easily identify the unimportant model parameters, facilitating the extraction of a submodel based on parameter values ranging from large to small until it aligns with a client's maximum computation capacity.

We further integrate this submodel construction method into our proposed FL algorithm FIARSE and comprehensively discuss how threshold-controlled biased gradient descent benefits the model-heterogeneity FL in the next section.

### 4.2 Algorithm Description

In FIARSE, a global model is initialized with arbitrary parameters $\tilde{\boldsymbol{x}}_0 \in \mathbb{R}^d$ (a pretrained model is allowed, which can be regarded as a special case of an arbitrary model).

Partial client participation is one of the features of FL algorithms because the server is unlikely to handle all the communications from all clients, especially when the number of clients is considerably large [10, 32, 44, 71, 74]. Therefore, we present our algorithm FIARSE in support of partial client participation: At the beginning of each communication round $t \in \{0, 1, \dots\}$, the server uniformly samples a group of clients from $[N]$ without replacement, denoted by $\mathcal{A}$, which consists of $A$ clients.

Subsequently, the server broadcasts the submodels to the selected clients and collects and merges their updates into the global model. In the rest of the section, we comprehensively discuss the details of these steps and exemplify them with $t$-th round.

**Submodel Extraction on Server.** Based on the set of participants $\mathcal{A}$, the server learns their computation capacities $\{\gamma_i\}_{i \in \mathcal{A}}$. Then, the server follows the threshold selection strategies described in Section 4.1 and extracts the submodel for all participants according to their computation capacities. Take the model-wise threshold selection as an example and select a submodel for participant $i \in \mathcal{A}$. The threshold is set for $\theta_i = \mathsf{TopK}_{\gamma_i}(|\tilde{\boldsymbol{x}}|)$. Then, the server extracts a submodel encompassing the parameters $|\tilde{\boldsymbol{x}}_t| \geq \theta_i$ and sends it to the participant. This procedure is equivalent to the expression in Line 3 of Algorithm 1, i.e., $\tilde{\boldsymbol{x}}_t \odot \mathcal{M}_t^{(i)}(\tilde{\boldsymbol{x}}_t)$.

**Local Training on Clients** $i \in \mathcal{A}$**.** After receiving the submodel from the server, we thereby initialize the local

---

**Algorithm 1** FIARSE

**Input:** local learning rate $\eta_l$, global learning rate $\eta_s$, local updates $K$, initial model $\tilde{\boldsymbol{x}}_0$.

1: **for** $t = 0, 1, 2, \ldots$ **do**
2:     Sample clients $\mathcal{A} \subseteq [N]$
3:     Send $\{\tilde{\boldsymbol{x}}_t \odot \mathcal{M}_t^{(i)}(\tilde{\boldsymbol{x}}_t)\}_{i \in \mathcal{A}}$ to clients $i \in \mathcal{A}$
4:     **for** $i \in \mathcal{A}$ **in parallel do**
5:         Initialize $\boldsymbol{x}_{t,0}^{(i)} = \tilde{\boldsymbol{x}}_t \odot \mathcal{M}_t^{(i)}(\tilde{\boldsymbol{x}}_t)$
6:         **for** $k = 0, \ldots, K-1$ **do**
7:             $g_{t,k+1}^{(i)} = \nabla_{\boldsymbol{x}_{t,k}^{(i)}} F_i \left( \boldsymbol{x}_{t,k}^{(i)} \odot \mathcal{M}_t^{(i)} \left( \boldsymbol{x}_{t,k}^{(i)} \right) \right)$
8:             $\boldsymbol{x}_{t,k+1}^{(i)} = \boldsymbol{x}_{t,k}^{(i)} - \eta_l \cdot g_{t,k+1}^{(i)}$
9:         **end for**
10:        $\Delta \boldsymbol{x}_t^{(i)} = \tilde{\boldsymbol{x}}_t - \boldsymbol{x}_{t,K}^{(i)}$
11:        Send $\Delta \boldsymbol{x}_t^{(i)}$ to the server
12:     **end for**
13:     $\tilde{\boldsymbol{x}}_{t+1} = \tilde{\boldsymbol{x}}_t - \eta_s \cdot \mathsf{Agg}_{i \in \mathcal{A}} \left( \Delta \boldsymbol{x}_t^{(i)} \right)$
14: **end for**

---

masking function $\mathcal{M}_t^{(i)}(\cdot)$ for training, which implicitly includes the threshold $\theta_i$. The threshold will remain constant during the local training. As outlined in Line $7-8$, the client utilizes the TCB-GD to optimize the local model for $K$ iterations. In each iteration $k \in \{0, \ldots, K-1\}$, the client utilizes the masking function to sort out the parameters that drop behind the threshold. Then, the client computes the gradient $\nabla_{\boldsymbol{x}_{t,k}^{(i)}} F_i \left( \boldsymbol{x}_{t,k}^{(i)} \odot \mathcal{M}_t^{(i)} \left( \boldsymbol{x}_{t,k}^{(i)} \right) \right)$ using Equation (3) (Line 7) and updates the local model via $\boldsymbol{x}_{t,k+1}^{(i)} = \boldsymbol{x}_{t,k}^{(i)} - \eta_l \cdot \nabla_{\boldsymbol{x}_{t,k}^{(i)}} F_i \left( \boldsymbol{x}_{t,k}^{(i)} \odot \mathcal{M}_t^{(i)} \left( \boldsymbol{x}_{t,k}^{(i)} \right) \right)$ (Line 8). As discussed in Section 4.1, the training memory is bounded by $O(\gamma_i d)$, and there is no additional storage requirement. In contrast to other importance-aware works such as Flado, the proposed FIARSE is more feasible in practice.

**Aggregation on Server.** After the selected clients finish their local updates, the server aggregates the updates from the clients (Line $10-11$). Similar to the global model aggregation of partial averaging described in Section 3, FIARSE updates the global model via $\tilde{\boldsymbol{x}}_{t+1} = \tilde{\boldsymbol{x}}_t - \eta_s \cdot \mathsf{Agg}_{i \in \mathcal{A}} \left( \Delta \boldsymbol{x}_t^{(i)} \right)$ (Line 13). A detailed description of the recursive function is placed in Appendix C.1.

Given that the global model starts with randomly initialized parameters, this aggregation not only updates the global model parameters but also aligns their values with their importance levels. In the next training round $t+1$, the FIARSE will return to Line 3, regenerating a submodel from the updated parameters, which is then sent back to the client. Since the values of the parameters represent better their importance levels than in the last training round, the newly generated submodel will contain more important parameters. The whole algorithm will then be repeated once again, resulting in a newly aggregated global model. This iterative process will progressively select important parameters and exclude unsignificant ones, steering the global model towards a state of convergence in which the importance of parameters is accurately represented. In the coming section, we theoretically analyze the convergence rate of the proposed FIARSE.

## 5 Convergence Analysis

Existing convergence analyses of FL algorithms predominantly rely on model-homogeneous settings [65]. However, the exploration of model-heterogeneity FL remains inadequately addressed, with some studies [58, 69, 76, 81] in this domain being recently introduced but relying on full client participation. This section aims to present a thorough convergence analysis of the proposed FIARSE under non-convex objectives. Specially, our analysis is established under the scenarios of model-heterogeneity FL where not all clients actively participate in the round-by-round training process.

Before showing the convergence result, we make the following three assumptions:

**Assumption 5.1** (Masked-$L$-smoothness). For all $i \in [N]$, the local objectives $F_i$ are $L$-Lipschitz smooth with a differentiable mask function $\mathcal{M}^{(i)}$: For all $w, v \in \mathbb{R}^d$,

$$\left\| \nabla_w F_i \left( w \odot \mathcal{M}^{(i)}(w) \right) - \nabla_v F_i \left( v \odot \mathcal{M}^{(i)}(v) \right) \right\|_2 \leq L \|w - v\|_2 .$$

**Assumption 5.2** (Bounded Global Variance). For all $i \in [N]$, the variance between local gradient $\nabla F_i(\cdot)$ and global gradient $\nabla F(\cdot)$ is bounded under the same model parameters: For all $w \in \mathbb{R}^d$, there exists a constant $\sigma_j \geq 0$ for all $j \in [n]$ such that

$$\sum_{i \in N_{\gamma'_j}} \frac{1}{\left| N_{\gamma'_j} \right|} \left\| \nabla F_i^{[\gamma'_{j-1}:\gamma'_j]}(w) - \nabla F^{[\gamma'_{j-1}:\gamma'_j]}(w) \right\|_2^2 \leq \sigma_j^2 .$$

We further annotate $\sigma^2 = \sum_{j=0}^n \sigma_j^2$.

**Assumption 5.3** (Masked Reduction). For all $i \in [N]$, the mask-incurred error is bounded with respect to the model parameter $\tilde{x}_t$, $t = 0, 1, \ldots$: There exists a scalar $\delta_t^2 \in [0,1)$ at round $t$ such that

$$\left\| \nabla F_i(\tilde{x}_t) \odot \mathcal{M}^{(i)}(\tilde{x}_t) - \nabla_{\tilde{x}_t} F_i \left( \tilde{x}_t \odot \mathcal{M}^{(i)}(\tilde{x}_t) \right) \right\|_2^2 \leq \delta_t^2 \|\tilde{x}_t\|_2^2 .$$

Lipschitz-smooth assumption has gained widespread acceptance in machine learning research, as evidenced by its incorporation in various studies such as [1, 6, 51, 70, 74, 75, 81]. Assumption 5.1 extends this assumption to a scenario where a binary mask is calculated based on the model parameters and subsequently applied to the model. The second assumption, widely made in the previous FL studies [74, 75], establishes bounds between the local objectives and the global objective due to the occurrence of non-i.i.d. data. Assumption 5.3 draws inspiration from [51, 81] and characterizes the masking performance by comparing gradients with and without the application of the mask. Notably, if $\mathcal{M}_t^{(i)} = \mathbf{1}^d$ (i.e., setting the threshold for $\theta_i = 0$), then $\delta_t^2 = 0$.

With these three assumptions, we analyze the convergence rate of our proposed algorithm. Under non-convex objectives, our goal is to evaluate if the gradient norm can approach zero with respect to the model parameters $\tilde{x}$ when the communication round $t \to \infty$. Theorem 5.4 presents the convergence result of the proposed FIARSE, and a detailed proof is deferred to Appendix C.

**Theorem 5.4.** *Suppose that Assumption 5.1, 5.2 and 5.3 hold. We define $F(\tilde{x}) \triangleq F\left(\tilde{x}, \mathbf{1}^{N \times d}\right)$, and $F(\tilde{x}) \geq F_*$ for all $\tilde{x} \in \mathbb{R}^d$. Let the local learning rate satisfy*

$$\eta_l \leq \min \left( \frac{1}{2L\sqrt{K(K+1)}}, \frac{1}{6L\sqrt{(K+1)A}}, \frac{\eta_s}{9L}, \frac{1}{12L\sqrt{KN}}, \frac{A}{32KNL\eta_s} \right).$$

*Denote $T$ as the total communication rounds. Therefore, the convergence rate of FIARSE for non-convex objectives should be*

$$\min_{t \in [T]} \|\nabla F(\tilde{x}_t)\|_2^2 \leq \frac{8\left(F(\tilde{x}_0) - F_*\right)N}{\eta_s \eta_l KAT} + \frac{64N}{A}\eta_s \eta_l KL\sigma^2 + \frac{32N}{T} \sum_{t \in [T]} \delta_t^2 \|\tilde{x}_t\|_2^2 . \tag{4}$$

The aforementioned theorem assesses the potential convergence of the global model towards a stable solution when employing the proposed algorithm (FIARSE). This implies that the submodels utilized in the analysis may differ from those employed in updates at each iteration. Consequently, our analysis takes into account clients utilizing the complete model, with the difference relative to their local submodels being constrained by Assumption 5.3. This methodology aligns with the analytical framework advocated by [81]. Next, we set the appropriate learning rates to achieve optimal convergence properties regarding the number of communication rounds, as outlined in the corollary:

**Corollary 5.5.** *Suppose that Assumption 5.1, 5.2 and 5.3 hold. We define $F(\tilde{x}) \triangleq F\left(\tilde{x}, \mathbf{1}^{N \times d}\right)$, and $F(\tilde{x}) \geq F_*$ for all $\tilde{x} \in \mathbb{R}^d$. Let the local learning rate $\eta_l = \frac{1}{K\sqrt{T}}$, and the global learning rate*

$\eta_s = 1$, *where $T$ is the total communication rounds. Then, under non-convex objectives, FIARSE converges to a small neighborhood of a stationary point of standard FL as $T$ is large enough, i.e.,*

$$\min_{t\in[T]} \|\nabla F\left(\tilde{\boldsymbol{x}}_t\right)\|_2^2 \leq O\left(\frac{N}{A} \cdot \frac{F(\tilde{\boldsymbol{x}}_0) - F_* + \sigma^2}{\sqrt{T}}\right) + O\left(\frac{N}{T} \sum_{t\in[T]} \delta_t^2 \|\tilde{\boldsymbol{x}}_t\|_2^2\right). \tag{5}$$

*where we treat $L$ as constants.*

*Remark* 5.6. Regarding the first term on the right-hand side (RHS) of Equation (5), it approaches zero as $T$ tends to infinity. However, an intriguing question arises concerning the potential for the second term to reach zero, given that the norm $\|\tilde{\boldsymbol{x}}_t\|_2^2$ cannot be zero. According to Assumption 5.3, a straightforward case where $\delta_t$ is always zero occurs when all clients use the full model – that is, when the threshold $\theta_i$ in equation Equation (3) is set to zero in the proposed FIARSE. Therefore, under model-homogeneous federated learning (FL), our theorem aligns with state-of-the-art works [34, 74] in terms of the convergence rate, which focuses solely on the number of communication rounds $T$, i.e., $O\left(1/\sqrt{T}\right)$.

For model-heterogeneous FL, it is noteworthy that $\delta_t$ may tend toward zero as $t \rightarrow \infty$, leading to $\frac{1}{T}\sum_{t=1}^T \delta_t^2 \|\tilde{\boldsymbol{x}}_t\|_2^2$ approaching zero. As explored by [51, 69, 81], this occurs because the submodels can replicate the performance of the full model on all clients after a substantial number of communication rounds $T$. Consequently, our proposed algorithm can achieve a convergence rate of $O\left(\frac{N\sigma^2}{A\sqrt{T}}\right)$, as long as $\lim_{t\rightarrow\infty} \delta_t = 0$.

Ignoring constant terms, the proposed FIARSE converges at a rate of $O\left(\sigma^2/\sqrt{T}\right)$ under full-client participation (i.e., $A = N$). Recently, some works have reported their convergence rates under model-heterogeneous FL with full-client participation, such as pruning-greedy [81] with a rate of $O\left(M\sigma^2/\sqrt{T}\right)$ and FedDSE [62] with $O\left(K\sigma^2/\sqrt{T}\right)$. Evidently, our proposed method exhibits better theoretical performance.

# 6 Experiments

## 6.1 Setup

**Datasets and Models.** We evaluate the proposed methodology using a combination of two computer vision (CV) datasets and one natural language processing (NLP) dataset. Specifically, we employ CIFAR-10 and CIFAR-100 datasets [37] for image classification, and the AGNews dataset [79] for text classification. For the first two datasets, we conduct training utilizing a ResNet-18 architecture [17], with modifications made by substituting its batch normalization (BN) layers with static BN counterparts [13]. For the AGNews dataset, we fine-tune a pre-trained RoBERTa-base model [48].

**Data Heterogeneity.** For CIFAR-10 and CIFAR-100, we follow [22, 30] and partition the datasets into 100 clients based on a Dirichlet distribution setting $\alpha = 0.3$. As for AGNews, we partition the datasets for 200 clients with Dirichlet distribution as well, but it is with the parameter of $\alpha = 1.0$. Note that the server or clients do not use any public datasets during the training stage. In the testing phase, we refer to the superset of all clients' test datasets as a "global test dataset."

**System Heterogeneity.** Specifically, the parameter $\gamma$ is defined as the ratio corresponding to the largest model that can be loaded onto a device. The experiments are conducted with four different model sizes represented by $\gamma' = \{1/64, 1/16, 1/4, 1.0\}$. The allocation of clients to each level is balanced. It's important to note that our proposed method is flexible and can accommodate varying numbers of complexity levels or client distributions.

**Implementation.** In this setting, we set the participation ratio to 10% by default. We perform 800-round training on the CV tasks while running for 300 rounds on the NLP task. To avoid the randomness of the results, we averaged the results from three different random seeds. In the experiments, we report the results of all the baselines based on the best hyperparameter settings. Due to the space limitation, more experimental results and analysis are deferred to Appendix D. Our code is released at `https://github.com/HarliWu/FIARSE`.

Table 1: Test accuracy under four different submodel sizes. To be more specific, the columns from "Local" to "Model (1.0)" evaluate the test accuracy on the local test datasets, while "Global" evaluates the average test accuracy of the global model of four different sizes (1/64, 1/16, 1/4, 1.0) on the global test dataset.

| Method | CIFAR-10 | | | | | | CIFAR-100 | | | | | | AGNews | |
|---|---|---|---|---|---|---|---|---|---|---|---|---|---|---|
| | Local | Model (1/64) | Model (1/16) | Model (1/4) | Model (1.0) | Global | Local | Model (1/64) | Model (1/16) | Model (1/4) | Model (1.0) | Global | Local | Global |
| HeteroFL | 68.88 | 60.24 | 69.32 | 72.18 | 73.76 | 66.05 | 31.75 | 27.24 | 29.80 | 33.52 | 36.44 | 30.67 | 87.59 | 86.88 |
| FedRolex | 67.18 | 54.60 | 64.96 | 70.08 | 79.08 | 65.98 | 31.67 | 21.00 | 30.84 | 36.44 | 38.40 | 29.89 | 87.43 | 87.19 |
| ScaleFL | 72.10 | 69.04 | 71.64 | 70.08 | 77.64 | 67.37 | 39.69 | 36.16 | 40.48 | 42.56 | 39.56 | 37.56 | 88.02 | 87.66 |
| FIARSE | **77.04** | **73.12** | **77.20** | **77.24** | **82.04** | **73.75** | **41.76** | **39.12** | **43.24** | **43.72** | **40.96** | **38.63** | **90.03** | **89.61** |

(a) HeteroFL Histogram   (b) FedRolex Histogram   (c) ScaleFL Histogram   (d) FIARSE Histogram

Figure 2: Histograms of various submodel extraction methods on CIFAR-10 under four submodel sizes. Each histogram shows the number of clients achieving different levels of test accuracy.

## 6.2 Submodel Performance on Local Dataset

In this setting, we evaluate the submodels' performance on each client's test datasets. To be specific, the local models are extracted from the global models. Figure 2 comprehensively illustrates the number of clients across different test accuracies. Among the four figures presented, all exhibit a left-skewed distribution, with the exception of FedRolex. This outcome aligns with our expectations, as FedRolex employs a rolling-based approach and is unable to concentrate on optimizing submodel performance on the local dataset, while the rest three approaches can spare efforts on a specific (HeteroFL and ScaleFL) or important (FIARSE) part, effectively addressing performance on local datasets. Among these three approaches, we notice that FIARSE stands out with the best results, showcasing superior performance as more clients achieve higher accuracy compared to the other two methods. The averaged results are also reported by Table 1 under the column of CIFAR-10 and "Local" to "Model (1.0)". Specifically, "Model (1/64)" to "Model (1.0)" shows the averaged local performances classified by the model sizes, and the "Local" shows the result by averaging across these four columns. Table 1 also presents the test accuracy of CIFAR-100 and AGNews. The proposed FIARSE achieves at least 2% better than other baselines under these datasets.

## 6.3 Submodel Performance on Global Dataset

In this setting, we evaluate the performance of submodels with various sizes on the global test dataset to assess the generalizability of our proposed algorithm. Table 1 presents the results of two CV datasets and one NLP dataset under the column "Global". In conjunction with Figure 3, our proposed method FIARSE constantly outperforms other baselines in all submodels with different sizes by a substantial margin. Figure 3 dives into the details of training and shows the test accuracy trend throughout the communication rounds. Consider the submodels are expected to surpass a 70% test accuracy threshold. As previously discussed, FIARSE ultimately achieves superior test accuracy compared to other baselines. Across model sizes of {1/64, 1/16, 1/4}, our proposed method requires fewer rounds to reach the targeted accuracy compared to other baselines. While the performance disparity between FIARSE and FedRolex is less discernible under the full model (Model (1.0)), both methods significantly outpace static submodel extraction approaches in achieving 70% accuracy. In summary, the proposed method stands out by attaining the desired submodels with the fewest rounds among the approaches implemented in this section.

## 6.4 Unparticipated Clients Performance

The above evaluations are conducted on the clients who participated in the training process. However, a more general scenario includes clients who skip the training phase but need models to process newly arrived data. In such cases, our algorithm can enable the server to customize models from the trained global model for them. Same as the expression in Line 3 of Algorithm 1, the server extracts a submodel encompassing the parameters $|\tilde{x}_t| \geq \theta_i$ and sends it to the client. Note that the unparticipated clients could have capacities different from that of any client involved in the training.

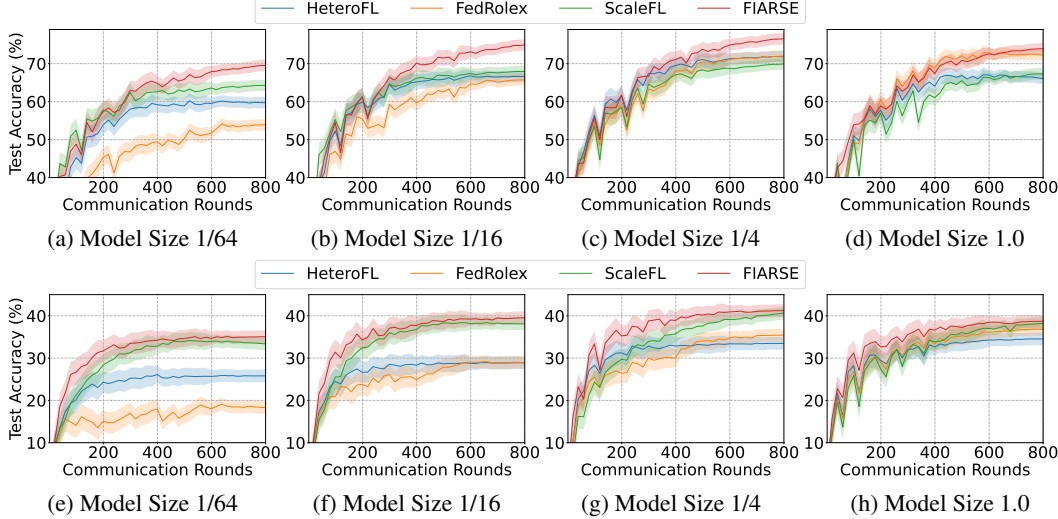

Figure 3: Comparison of test accuracy across communication rounds for different submodel extraction strategies under four varying model sizes (1/64, 1/16, 1/4, 1.0) on global test datasets of CIFAR-10 (upper, a – d) and CIFAR-100 (lower, e – h).

In Figure 4, we employ our algorithm as well as the baselines to extract submodels with different sizes and compare their performance on unparticipated clients. Generally, the performances of all methods increase as the size of extracted submodels grows. Our algorithm consistently outperforms the baseline methods. In contrast, alternative approaches, particularly static ones like HeteroFL and ScaleFL, suffer from more significant drops in performance. For the case of extracting a submodel that is much smaller than the minimum size involved in the training stage, our proposed method demonstrates remarkable superiority, outperforming existing works by at least 10%.

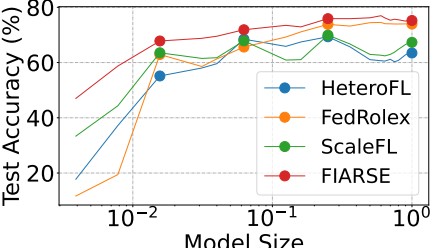

Figure 4: Comparison of test accuracy across different submodel sizes for different submodel extraction methods on a global test dataset of CIFAR-10.

## 7 Conclusion

This work introduces an algorithm for model-heterogeneity FL, named FIARSE, that utilizes importance-aware operation to extract various sizes of submodels. In detail, we utilize TCB-GD that is able to optimize the clients' local parameters to reflect their importance levels. Subsequently, we provide a theoretical analysis and highlight that the proposed work can converge to a neighborhood of a stationary point at the rate of $O\left(1/\sqrt{T}\right)$, where $T$ is the number of communication rounds. Extensive experiments are conducted on ResNet-18 and Roberta-base that demonstrate the significant superiority of our proposed method over the state-of-the-art works.

The proposed approach relies on exploiting model sparsity, which is conditionally supported by some hardware. In light of this limitation, one of the future works is to investigate neuron-wise importance-aware submodel extraction, a method that calculates the importance level of neurons without depending on additional information.

## Broader Impact

This work addresses model heterogeneity in federated learning due to varying computational capacities among clients. The proposed method enhances the efficiency of on-device training and reduces computation and energy demands, which is particularly significant for resource-constrained devices like smartphones in real-world applications. Moreover, the method facilitates the practical deployment of federated learning systems in heterogeneous environments, making them more accessible and scalable.

## Acknowledgements

The authors would like to thank the anonymous reviewers for their constructive comments. This work is supported in part by the US National Science Foundation under grants NSF IIS-1747614, NSF IIS-2226108, and NSF CNS-2154059. Any opinions, findings, and conclusions or recommendations expressed in this material are those of the author(s) and do not necessarily reflect the views of the National Science Foundation.

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

# A    Related Work

**Computation Heterogeneity in FL.**    Computation heterogeneity in FL has become a critical area of research due to the varying computational capacities of participating clients. In traditional FL setups, each client is expected to perform the same volume of computation, such as training on local datasets for a fixed number of iterations, regardless of their hardware capabilities or resource availability [14, 15, 18, 24, 63, 68]. This assumption leads to inefficiencies, particularly when slower devices cause stragglers that delay model aggregation [23, 44, 64]. Recent work has proposed adaptive aggregation strategies to handle computation heterogeneity by assigning different workloads or varying the number of local updates based on client resources, allowing faster clients to contribute more to the global model while slower clients contribute less frequently [43, 50, 53, 57, 64, 72]. However, these works require the clients to train the full model, so they are infeasible when some of the clients cannot load the full model due to limited computation resources. In contrast, our work extracts a submodel for each client that fits within their computational capacity.

**Model Customization in FL.**    In addition to submodel extraction, another effective method for facilitating model heterogeneity is model customization [11, 28, 46, 66, 77, 80]. This approach allows clients to construct local models that align with their local computation resources. Since the clients' local models could be very distinct, the existing works implement model aggregation through the technique of knowledge distillation [19, 46, 61, 78]. In detail, each client first locally trains its model on a shared public dataset, and then sends the model's output (i.e., logits) to the server. The server gathers and aggregates clients' outputs into a unified model output and broadcasts it to the clients [60]. Prior to the next round of local training, the clients thereby fine-tune the local models by incorporating the aggregated logits. Although the process effectively transfers knowledge across heterogeneous models, model customization requires a shared public dataset that should have a similar distribution with the clients' training data, which is unattainable in some cases [3, 59]. In our work, we do not require a public dataset, rendering it applicable to a broader range of applications.

**Model Sparsification in FL.**    Model sparsification, also known as model pruning, intrigues increasing research focuses with the introduction of lottery ticket hypothesis [16]. This hypothesis addresses the existence of a sparse submodel within a large model, capable of direct training to yield improved performance. Early works [9, 47, 82] have explored sparse models to mitigate the computation overhead. Notably, the industry has recently achieved remarkable advancements in the development of hardware that facilitates the training and inference of sparse models [20, 26, 38, 39].

In the context of FL, two distinct strategies in search of an optimal sparse submodel: dense-to-sparse [27, 31, 41] and sparse-to-sparse [5, 12, 40, 55, 56]. These strategies are distinguished based on whether the global model starts at dense. In both approaches, clients initialize with a global model and iteratively train towards a sparse model to mitigate over-parameterization. Nonetheless, these methods face challenges when client computation capacities differ, particularly when loading the global model exceeds the maximum computation capacities of some clients. Consequently, there arises a need for algorithms capable of tailoring submodels to accommodate various client computation capacities.

An existing work, Flado [45], realizes that the server tailors the submodels to align with various computation resources among clients. It achieves the performance trade-off between the global model and the clients' submodels by differentiating model parameters based on their importance levels. However, this advantage comes at the cost of requiring clients to explicitly maintain the importance score of each model parameter. This requirement, consequently, introduces a considerable overhead on each client in terms of both storage and computation, which imposes great challenges on resource-constrained devices. Contrasting with Flado, our FIARSE explores the correlation between a model parameter's importance level and its value. It implicitly represents the importance level through the parameter's value, thereby eliminating the need to explicitly maintain separate importance scores for each model parameter.

## B    Derivation of Equation (3)

Before deriving Equation (3), we should find a formula for $\frac{\partial \mathcal{M}^{(i)}}{\partial \tilde{x}}$:

$$\frac{\partial \mathcal{M}^{(i)}}{\partial \tilde{x}} = \bigcup_j \frac{\partial}{\partial \tilde{x}_j} \left( \frac{|\tilde{x}_j| - \theta_i}{|\tilde{x}_j| + \theta_i} \right) \cdot \mathbf{1}_{|\tilde{x}_j| \geq \theta_i} \tag{6}$$

$$= \bigcup_j \frac{\partial |\tilde{x}_j|}{\partial x_j} \cdot \frac{\partial}{\partial |\tilde{x}_j|} \left( \frac{|\tilde{x}_j| - \theta_i}{|\tilde{x}_j| + \theta_i} \right) \cdot \mathbf{1}_{|\tilde{x}_j| \geq \theta_i} \tag{7}$$

$$= \bigcup_j \frac{\partial |\tilde{x}_j|}{\partial x_j} \cdot \frac{2\theta_i}{(|\tilde{x}_j| + \theta_i)^2} \cdot \mathbf{1}_{|\tilde{x}_j| \geq \theta_i} \tag{8}$$

As defined in the paper, $\mathcal{M}^{(i)} = \cup_j \mathbf{1}_{|\tilde{x}_j| \geq \theta_i}$. Therefore, we can obtain Equation (3) via

$$\frac{\partial F_i(\tilde{x} \odot \mathcal{M}^{(i)})}{\partial \tilde{x}} = \frac{\partial F_i(\tilde{x} \odot \mathcal{M}^{(i)})}{\partial \tilde{x} \odot \mathcal{M}^{(i)}} \cdot \frac{\partial \tilde{x} \odot \mathcal{M}^{(i)}}{\partial \tilde{x}} \tag{9}$$

$$= \frac{\partial F_i(\tilde{x} \odot \mathcal{M}^{(i)})}{\partial \tilde{x} \odot \mathcal{M}^{(i)}} \odot \left( \mathcal{M}^{(i)} + \tilde{x} \odot \frac{\partial \mathcal{M}^{(i)}}{\partial \tilde{x}} \right) \tag{10}$$

$$= \nabla F_i(\tilde{x} \odot \mathcal{M}^{(i)}) \odot \mathcal{M}^{(i)} \left( 1 + \frac{2|\tilde{x}|\theta_i}{(|\tilde{x}| + \theta_i)^2} \right) \tag{11}$$

where $\odot$ is an element-wise product.

## C    Proof of Theorem 5.5

Prior to giving detailed proofs of the theorems, we cover some technical lemmas in this section, and all of them are valid in general cases.

### C.1    Useful Lemmas

**Comprehensive description for the aggregation on the server.**    Since FIARSE generates the submodel based on the parameter importance reflected by the magnitude of the model parameters, a submodel is nested within other submodels that are with a larger size. Given a set of model sizes $\{\gamma_i\}_{i \in [N]}$, we define a sorted set $\gamma' = \cup_{i \in [N]} \{\gamma_i\}$, where $0 < \gamma'_0 < \cdots < \gamma'_{n-1} \leq 1$ and $n = |\gamma'| \leq N$. Let us partition the global model $\tilde{x}_t$ into the disjoint submodels $\tilde{x}_{t,[0:\gamma'_0]}$, $\tilde{x}_{t,[\gamma'_0:\gamma'_1]}$, $\ldots$, $\tilde{x}_{t,[\gamma'_{n-2}:\gamma'_{n-1}]}$, $\tilde{x}_{t,[\gamma'_{n-1}:1]}$. These models are separately held by a set of clients $N_{\gamma'_0} \supset N_{\gamma'_1} \supset \cdots \supset N_{\gamma'_{n-1}} \supset \emptyset$. In round $t$, the server samples a set of clients $\mathcal{A}$ to train the model, and these models are therefore held by $\mathcal{A}_{\gamma'_0} \supseteq \mathcal{A}_{\gamma'_1} \supseteq \cdots \supseteq \mathcal{A}_{\gamma'_{n-1}} \supseteq \emptyset$. Notably, the smallest submodel should be held by all clients, i.e., $N_{\gamma'_0} = [N]$, $\mathcal{A}_{\gamma'_0} = \mathcal{A}$. Let us define $\mathsf{Agg}_{i \in \mathcal{A}} \left( \Delta \boldsymbol{x}_t^{(i)} \right) = \cup_j \bar{\Delta} \tilde{\boldsymbol{x}}_{t,[\gamma'_{j-1}:\gamma'_j]}$, where

$$\bar{\Delta} \tilde{\boldsymbol{x}}_{t,[\gamma'_{j-1}:\gamma'_j]} = \begin{cases} \frac{1}{\left| \mathcal{A}_{\gamma'_j} \right|} \sum_{i \in \mathcal{A}_{\gamma'_j}} \Delta \boldsymbol{x}_{t,[\gamma'_{j-1}:\gamma'_j]}^{(i)}, & \left| \mathcal{A}_{\gamma'_j} \right| \neq 0 \\ \mathbf{0}, & \left| \mathcal{A}_{\gamma'_j} \right| = 0 \end{cases} \tag{12}$$

Therefore, the global update follows $\tilde{x}_{t+1} = \tilde{x}_t - \eta_s \mathsf{Agg}_{i \in \mathcal{A}} \left( \tilde{x}_t - \boldsymbol{x}_{t,K}^{(i)} \right)$. Upon the description, we have the following lemma:

**Lemma C.1.** *For a vector $v \in \mathbb{R}^d$, we partition it into $n$ sets, where we present it in Table 2. Suppose we select an $A$-client set out of a total of $N$ clients $[N]$. Let us define that*

$$\mathsf{Agg}_{i \in [N]} (v) = \bigcup_{j \in [n]} \frac{1}{\left| N_{\gamma'_j} \right|} \sum_{i \in N_{\gamma'_j}} v_{[\gamma'_{j-1}:\gamma'_j]}^{(i)}; \quad \mathsf{Agg}_{i \in \mathcal{A}} (v) = \bigcup_{j \in [n]} \frac{1}{\left| \mathcal{A}_{\gamma'_j} \right|} \sum_{i \in \mathcal{A}_{\gamma'_j}} v_{[\gamma'_{j-1}:\gamma'_j]}^{(i)}. \tag{13}$$

Table 2: Vector Partition across Different Sizes. These notations are the same as the definition in Equation (12).

| Submodel | $v_{[0:\gamma_0']}$ | $v_{[\gamma_0':\gamma_1']}$ | $\cdots$ | $v_{[\gamma_{n-2}':\gamma_{n-1}']}$ | $v_{[\gamma_{n-1}':1]}$ |
|---|---|---|---|---|---|
| Clients set | $N_{\gamma_0'}$ | $N_{\gamma_1'}$ | $\ldots$ | $N_{\gamma_{n-1}'}$ | $\emptyset$ |
| Participation set | $\mathcal{A}_{\gamma_0'}$ | $\mathcal{A}_{\gamma_1'}$ | $\ldots$ | $\mathcal{A}_{\gamma_{n-1}'}$ | $\emptyset$ |

*Therefore, we have*

$$\mathbb{E}\textbf{Agg}_{i\in\mathcal{A}}(v) = \bigcup_{j\in[n]} \frac{1}{\left|N_{\gamma_j'}\right|} \cdot \frac{\binom{N}{A} - \binom{N-\left|N_{\gamma_j'}\right|}{A}}{\binom{N}{A}} \sum_{i\in N_{\gamma_j'}} v^{(i)}_{[\gamma_{j-1}':\gamma_j']} \tag{14}$$

*Specially, if $A > N - \left|N_{\gamma_j'}\right|$, we define $\binom{N-\left|N_{\gamma_j'}\right|}{A} = 0$.*

*Proof.* For a given submodel $v_{[\gamma_{j-1}':\gamma_j']}$, there are $\left|N_{\gamma_j'}\right|$ clients holding the submodel. In other words, $\left(N - \left|N_{\gamma_j'}\right|\right)$ clients do not hold any parameters of $v_{[\gamma_{j-1}':1]}$. If a selected client $i \in \mathcal{A}$ holds the submodel $v_{[\gamma_{j-1}':\gamma_j']}$, it will be divided by an integer $k \in \{1, \ldots, A\}$ (i.e., $\left|\mathcal{A}_{\gamma_j'}\right| = k$) with a probability of $\frac{\binom{\left|N_{\gamma_j'}\right|-1}{k-1} \cdot \binom{N-\left|N_{\gamma_j'}\right|}{A-k}}{\binom{N}{A}}$. Therefore, this part is expected to obtain the following result after aggregation:

$$\mathbb{E}\textbf{Agg}_{i\in\mathcal{A}}\left(v_{[\gamma_{j-1}':\gamma_j']}\right) = \sum_{k=1}^{A} \frac{\binom{\left|N_{\gamma_j'}\right|-1}{k-1} \cdot \binom{N-\left|N_{\gamma_j'}\right|}{A-k}}{\binom{N}{A}} \cdot \frac{1}{k} \sum_{i\in N_{\gamma_j'}} v^{(i)}_{[\gamma_{j-1}':\gamma_j']} \tag{15}$$

$$= \sum_{k=1}^{A} \frac{\binom{\left|N_{\gamma_j'}\right|}{k} \cdot \binom{N-\left|N_{\gamma_j'}\right|}{A-k}}{\binom{N}{A}} \cdot \frac{1}{\left|N_{\gamma_j'}\right|} \sum_{i\in N_{\gamma_j'}} v^{(i)}_{[\gamma_{j-1}':\gamma_j']} \tag{16}$$

$$= \frac{\binom{N}{A} - \binom{N-\left|N_{\gamma_j'}\right|}{A}}{\binom{N}{A}} \cdot \frac{1}{\left|N_{\gamma_j'}\right|} \sum_{i\in N_{\gamma_j'}} v^{(i)}_{[\gamma_{j-1}':\gamma_j']} \tag{17}$$

By concatenating all $j \in [n]$, we can attain the desired results. $\qquad\square$

In addition to the above lemma, we provide a bound for $\frac{\binom{N}{A}-\binom{N-c}{A}}{\binom{N}{A}}$ and $\frac{\binom{N}{A}-\binom{N-c}{A}}{c\cdot\binom{N}{A}}$, where $c$ is an integer of $\{1, \ldots, N-A\}$.

**Lemma C.2.** *Given $N$ and $A$ are two integers where $A \le N$, and $c$ is an integer of $\{1, \ldots, N-A\}$, the following two inequalities are always true:*

$$\frac{A}{N} \le \frac{\binom{N}{A}-\binom{N-c}{A}}{\binom{N}{A}} \le 1; \qquad \frac{\binom{N}{A}-\binom{N-c}{A}}{c\cdot\binom{N}{A}} \le \frac{A}{N} \tag{18}$$

*Proof.* For the first inequality, the lower bound holds when $c = 1$, while the upper bound constantly satisfies.

For the second inequality, we define $f(c) = \frac{\binom{N}{A}-\binom{N-c}{A}}{c\cdot\binom{N}{A}}$, and we aim to show that $f(c)$ is *monotonically decreasing.*

Toward the goal, we reduce and prove the following inequality holds:

$$\frac{\binom{N}{A} - \binom{N-c}{A}}{c \cdot \binom{N}{A}} \leq \frac{\binom{N}{A} - \binom{N-(c-1)}{A}}{(c-1) \cdot \binom{N}{A}} \tag{19}$$

$$\Longleftrightarrow \quad c \cdot \binom{N-c+1}{A} - (c-1) \cdot \binom{N-c}{A} \leq \binom{N}{A} \tag{20}$$

$$\Longleftrightarrow \quad \frac{(N-c)!}{(N-c-A+1)!} \cdot (c(N-c+1) - (c-1)(N-c-A+1)) \leq \frac{N!}{(N-A)!A!} \tag{21}$$

$$\Longleftrightarrow \quad N + cA - c - A + 1 \leq (N-c-A+1) \cdot \frac{\frac{N!}{(N-c)!}}{\frac{(N-A)!}{(N-A-c)!}} \tag{22}$$

$$\Longleftrightarrow \quad 1 + \frac{cA}{N-c-A+1} \leq \prod_{i=0}^{c-1} \left(1 + \frac{A}{N-A-i}\right) \tag{23}$$

Next, we show the establishment of Equation (23) by mathematical induction:

• When $c = 1$, we have $LHS = 1 + \frac{A}{N-A}$, while $RHS = 1 + \frac{A}{N-A}$. Apparently, the inequality holds.

• Let us assume that, when $c = k$, the conclusion holds, i.e., $1 + \frac{kA}{N-k-A+1} \leq \prod_{i=0}^{k-1} \left(1 + \frac{A}{N-A-i}\right)$. When $c = k+1$, we have

$$\prod_{i=0}^{k} \left(1 + \frac{A}{N-A-i}\right) = \prod_{i=0}^{k-1} \left(1 + \frac{A}{N-A-i}\right) \cdot \left(1 + \frac{A}{N-A-k}\right) \tag{24}$$

$$\geq \left(1 + \frac{kA}{N-k-A+1}\right) \cdot \left(1 + \frac{A}{N-A-k}\right) \tag{25}$$

$$\geq 1 + \frac{(k+1)A}{N-k-A} \tag{26}$$

This shows that the inequality (i.e., Equation (23)) still holds, indicating that $f(c)$ is monotonically decreasing. As $c$ is an integer of $\{1, \ldots, N - A\}$, we have

$$f(c) \leq f(1) = \frac{\binom{N}{A} - \binom{N-1}{A}}{\binom{N}{A}} = \frac{A}{N} \tag{27}$$

Apparently, we can attain the desired results as mentioned in Equation (18). $\qquad\square$

## C.2 Preliminary

**Lemma C.3.** *Suppose that Assumption 5.1, 5.2 and 5.3 hold. Let the local learning rate satisfy $\eta_l \leq \frac{1}{2L\sqrt{K(K+1)}}$. With FIARSE, we have the following conclusion:*

$$\frac{1}{N} \sum_{i \in [N]} \sum_{k=0}^{K-1} \mathbb{E} \left\| x_{t,k}^{(i)} - \tilde{x}_t \right\|_2^2 \leq 36K^2 \eta_l^2 \left(\delta_t^2 \|\tilde{x}_t\|_2^2 + \sigma^2 + \|\nabla F(\tilde{x}_t)\|_2^2\right) \tag{28}$$

*Proof.* First, we establish a recursive relationship between $\mathbb{E} \left\| x_{t,k+1}^{(i)} - \tilde{x}_t \right\|_2^2$ and $\mathbb{E} \left\| x_{t,k}^{(i)} - \tilde{x}_t \right\|_2^2$:

$$\mathbb{E} \left\| x_{t,k+1}^{(i)} - \tilde{x}_t \right\|_2^2 = \mathbb{E} \left\| x_{t,k}^{(i)} - \eta_l \nabla_{x_{t,k}^{(i)}} F_i \left(x_{t,k}^{(i)} \odot \mathcal{M}_t^{(i)} \left(x_{t,k}^{(i)}\right)\right) - \tilde{x}_t \right\|_2^2 \tag{29}$$

$$\leq \left(1 + \frac{1}{K}\right) \cdot \mathbb{E} \left\| x_{t,k}^{(i)} - \tilde{x}_t \right\|_2^2 + (1+K)\eta_l^2 \cdot \mathbb{E} \left\| \nabla_{x_{t,k}^{(i)}} F_i \left(x_{t,k}^{(i)} \odot \mathcal{M}_t^{(i)} \left(x_{t,k}^{(i)}\right)\right) \right\|_2^2 \tag{30}$$

$$\leq \left(1 + \frac{1}{K}\right) \cdot \mathbb{E} \left\| x_{t,k}^{(i)} - \tilde{x}_t \right\|_2^2 \tag{31}$$

$$+ 4(1+K)\eta_l^2 \cdot \mathbb{E} \left\| \nabla_{\boldsymbol{x}_{t,k}^{(i)}} F_i \left( \boldsymbol{x}_{t,k}^{(i)} \odot \mathcal{M}_t^{(i)} \left( \boldsymbol{x}_{t,k}^{(i)} \right) \right) - \nabla_{\tilde{\boldsymbol{x}}_t} F_i \left( \tilde{\boldsymbol{x}}_t \odot \mathcal{M}_t^{(i)} \left( \tilde{\boldsymbol{x}}_t \right) \right) \right\|_2^2 \quad (32)$$

$$+ 4(1+K)\eta_l^2 \cdot \mathbb{E} \left\| \nabla_{\tilde{\boldsymbol{x}}_t} F_i \left( \tilde{\boldsymbol{x}}_t \odot \mathcal{M}_t^{(i)} \left( \tilde{\boldsymbol{x}}_t \right) \right) - \nabla_{\tilde{\boldsymbol{x}}_t} F_i \left( \tilde{\boldsymbol{x}}_t \right) \odot \mathcal{M}_t^{(i)} \left( \tilde{\boldsymbol{x}}_t \right) \right\|_2^2 \quad (33)$$

$$+ 4(1+K)\eta_l^2 \cdot \mathbb{E} \left\| \nabla_{\tilde{\boldsymbol{x}}_t} F_i \left( \tilde{\boldsymbol{x}}_t \right) \odot \mathcal{M}_t^{(i)} \left( \tilde{\boldsymbol{x}}_t \right) - \nabla F \left( \tilde{\boldsymbol{x}}_t \right) \odot \mathcal{M}_t^{(i)} \left( \tilde{\boldsymbol{x}}_t \right) \right\|_2^2 \quad (34)$$

$$+ 4(1+K)\eta_l^2 \cdot \mathbb{E} \left\| \nabla F \left( \tilde{\boldsymbol{x}}_t \right) \odot \mathcal{M}_t^{(i)} \left( \tilde{\boldsymbol{x}}_t \right) \right\|_2^2 \quad (35)$$

where the second term in Equation (30) is expanded to four parts: Equation (32), Equation (33), Equation (34), and Equation (35). Furthermore, applying Assumption 5.1 to Equation (32) and Assumption 5.3 to Equation (33), along with the equalities $\nabla_{\tilde{\boldsymbol{x}}_t} F_i \left( \tilde{\boldsymbol{x}}_t \right) \odot \mathcal{M}_t^{(i)} \left( \tilde{\boldsymbol{x}}_t \right) = \nabla F_i^{[0:\gamma_i]} \left( \tilde{\boldsymbol{x}}_t \right)$ and $\nabla F \left( \tilde{\boldsymbol{x}}_t \right) \odot \mathcal{M}_t^{(i)} \left( \tilde{\boldsymbol{x}}_t \right) = \nabla F^{[0:\gamma_i]} \left( \tilde{\boldsymbol{x}}_t \right)$, we simplify the above inequality for

$$\mathbb{E} \left\| \boldsymbol{x}_{t,k+1}^{(i)} - \tilde{\boldsymbol{x}}_t \right\|_2^2 \quad (36)$$

$$\leq \left( 1 + \frac{1}{K} + 4(1+K)\eta_l^2 L^2 \right) \cdot \mathbb{E} \left\| \boldsymbol{x}_{t,k}^{(i)} - \tilde{\boldsymbol{x}}_t \right\|_2^2 \quad (37)$$

$$+ 4(1+K)\eta_l^2 \left( \delta_t^2 \left\| \tilde{\boldsymbol{x}}_t \right\|_2^2 + \mathbb{E} \left\| \nabla F_i^{[0:\gamma_i]} \left( \tilde{\boldsymbol{x}}_t \right) - \nabla F^{[0:\gamma_i]} \left( \tilde{\boldsymbol{x}}_t \right) \right\|_2^2 + \mathbb{E} \left\| \nabla F^{[0:\gamma_i]} \left( \tilde{\boldsymbol{x}}_t \right) \right\|_2^2 \right) \quad (38)$$

Let $C$ be

$$C = \delta_t^2 \left\| \tilde{\boldsymbol{x}}_t \right\|_2^2 + \mathbb{E} \left\| \nabla F_i^{[0:\gamma_i]} \left( \tilde{\boldsymbol{x}}_t \right) - \nabla F^{[0:\gamma_i]} \left( \tilde{\boldsymbol{x}}_t \right) \right\|_2^2 + \mathbb{E} \left\| \nabla F^{[0:\gamma_i]} \left( \tilde{\boldsymbol{x}}_t \right) \right\|_2^2. \quad (39)$$

As $\eta_l \leq \frac{1}{2L\sqrt{K(K+1)}}$, we have

$$\mathbb{E} \left\| \boldsymbol{x}_{t,k+1}^{(i)} - \tilde{\boldsymbol{x}}_t \right\|_2^2 \leq \left( 1 + \frac{2}{K} \right) \mathbb{E} \left\| \boldsymbol{x}_{t,k}^{(i)} - \tilde{\boldsymbol{x}}_t \right\|_2^2 + 4(1+K)\eta_l^2 \cdot C \quad (40)$$

$$\leq 4(1+K) \left( 1 + \frac{2}{K} \right)^k \eta_l^2 \cdot C \quad (41)$$

$$\leq 36(1+K)\eta_l^2 \cdot C \quad (42)$$

Equation (42) holds because $\left( 1 + \frac{2}{K} \right)^k \leq 9$ for all $k \in \mathbb{N}_+$. Therefore, for all $k \in [K]$, we have

$$\mathbb{E} \left\| \boldsymbol{x}_{t,k}^{(i)} - \tilde{\boldsymbol{x}}_t \right\|_2^2 \leq 36(1+K)\eta_l^2 \delta_t^2 \left\| \tilde{\boldsymbol{x}}_t \right\|_2^2 \quad (43)$$

$$+ 36(1+K)\eta_l^2 \left( \mathbb{E} \left\| \nabla F_i^{[0:\gamma_i]} \left( \tilde{\boldsymbol{x}}_t \right) - \nabla F^{[0:\gamma_i]} \left( \tilde{\boldsymbol{x}}_t \right) \right\|_2^2 + \mathbb{E} \left\| \nabla F^{[0:\gamma_i]} \left( \tilde{\boldsymbol{x}}_t \right) \right\|_2^2 \right). \quad (44)$$

With this conclusion, we can bound the left-hand side of Equation (28), which is

$$\frac{1}{N} \sum_{i \in [N]} \sum_{k=0}^{K-1} \mathbb{E} \left\| \boldsymbol{x}_{t,k}^{(i)} - \tilde{\boldsymbol{x}}_t \right\|_2^2 = \frac{1}{N} \sum_{i \in [N]} \sum_{k=1}^{K-1} \mathbb{E} \left\| \boldsymbol{x}_{t,k}^{(i)} - \tilde{\boldsymbol{x}}_t \right\|_2^2 \quad (45)$$

$$\leq \frac{36K^2\eta_l^2}{N} \sum_{i \in [N]} \left( \delta_t^2 \left\| \tilde{\boldsymbol{x}}_t \right\|_2^2 + \mathbb{E} \left\| \nabla F_i^{[0:\gamma_i]} \left( \tilde{\boldsymbol{x}}_t \right) - \nabla F^{[0:\gamma_i]} \left( \tilde{\boldsymbol{x}}_t \right) \right\|_2^2 + \mathbb{E} \left\| \nabla F^{[0:\gamma_i]} \left( \tilde{\boldsymbol{x}}_t \right) \right\|_2^2 \right) \quad (46)$$

Finally, by applying Assumption 5.2 to Equation (46), we achieve the desired conclusion. $\square$

**Lemma C.4.** *Suppose that Assumption 5.1, 5.2 and 5.3 hold. Let the local learning rate satisfy* $\eta_l \leq \min \left( \frac{1}{2L\sqrt{K(K+1)}}, \frac{1}{6L\sqrt{(K+1)A}} \right)$. *For any* $t \in [T]$, $j \in \{0, \ldots, n\}$, *let* $\boldsymbol{v}_t^{[\gamma'_{j-1}:\gamma'_j]}$ *be*

$$\boldsymbol{v}_t^{[\gamma'_{j-1}:\gamma'_j]} = \nabla F^{[\gamma'_{j-1}:\gamma'_j]} \left( \tilde{\boldsymbol{x}}_t \right) - \frac{1}{\left| N_{\gamma'_j} \right| K} \sum_{i \in N_{\gamma'_j}} \sum_{k=0}^{K-1} \nabla_{\boldsymbol{x}_{t,k}^{(i)}} F_i^{[\gamma'_{j-1}:\gamma'_j]} \left( \boldsymbol{x}_{t,k}^{(i)} \odot \mathcal{M}_t^{(i)} \left( \boldsymbol{x}_{t,k}^{(i)} \right) \right) \quad (47)$$

*With FIARSE, we have the following conclusion:*

$$\sum_{j=0}^{n} \frac{\binom{N}{A} - \binom{N - \left|N_{\gamma_j'}\right|}{A}}{\binom{N}{A}} \cdot \mathbb{E}\left\|\boldsymbol{v}_t^{[\gamma_{j-1}':\gamma_j']}\right\|_2^2 \leq 4A\delta_t^2 \left\|\tilde{\boldsymbol{x}}_t\right\|_2^2 + 72KA\eta_l^2 L^2 \left(\sigma^2 + \left\|\nabla F\left(\tilde{\boldsymbol{x}}_t\right)\right\|_2^2\right) \quad (48)$$

*Proof.* In the beginning, let us define

$$\hat{\boldsymbol{g}}_t^{[\gamma_{j-1}':\gamma_j'](i)} = \nabla F_i^{[\gamma_{j-1}':\gamma_j']}\left(\tilde{\boldsymbol{x}}_t\right) - \nabla_{\tilde{\boldsymbol{x}}_t} F_i^{[\gamma_{j-1}':\gamma_j']}\left(\tilde{\boldsymbol{x}}_t \odot \mathcal{M}_t^{(i)}\left(\tilde{\boldsymbol{x}}_t\right)\right), \quad (49)$$

$$\tilde{\boldsymbol{g}}_{t,k}^{[\gamma_{j-1}':\gamma_j'](i)} = \nabla_{\boldsymbol{x}_{t,k}^{(i)}} F_i^{[\gamma_{j-1}':\gamma_j']}\left(\boldsymbol{x}_{t,k}^{(i)} \odot \mathcal{M}_t^{(i)}\left(\boldsymbol{x}_{t,k}^{(i)}\right)\right) - \nabla_{\tilde{\boldsymbol{x}}_t} F_i^{[\gamma_{j-1}':\gamma_j']}\left(\tilde{\boldsymbol{x}}_t \odot \mathcal{M}_t^{(i)}\left(\tilde{\boldsymbol{x}}_t\right)\right). \quad (50)$$

According to Equation (47), $\boldsymbol{v}^{[\gamma_{j-1}':\gamma_j']}$ is equivalent to

$$\boldsymbol{v}_t^{[\gamma_{j-1}':\gamma_j']} = \frac{1}{\left|N_{\gamma_j'}\right|} \sum_{i \in N_{\gamma_j'}} \hat{\boldsymbol{g}}_t^{[\gamma_{j-1}':\gamma_j'](i)} - \frac{1}{\left|N_{\gamma_j'}\right| K} \sum_{i \in N_{\gamma_j'}} \sum_{k=0}^{K-1} \tilde{\boldsymbol{g}}_{t,k}^{[\gamma_{j-1}':\gamma_j'](i)} \quad (51)$$

To find a bound for Equation (48), we try to bound $\mathbb{E}\left\|\boldsymbol{v}\right\|_2^2$ using generalized mean inequality and Cauchy–Shwarz inequality:

$$\mathbb{E}\left\|\boldsymbol{v}_t^{[\gamma_{j-1}':\gamma_j']}\right\|_2^2 \leq 2\mathbb{E}\left\|\frac{1}{\left|N_{\gamma_j'}\right|} \sum_{i \in N_{\gamma_j'}} \hat{\boldsymbol{g}}_t^{[\gamma_{j-1}':\gamma_j'](i)}\right\|_2^2 + 2\mathbb{E}\left\|\frac{1}{\left|N_{\gamma_j'}\right| K} \sum_{i \in N_{\gamma_j'}} \sum_{k=0}^{K-1} \tilde{\boldsymbol{g}}_{t,k}^{[\gamma_{j-1}':\gamma_j'](i)}\right\|_2^2$$
$$(52)$$

$$\leq 2\frac{1}{\left|N_{\gamma_j'}\right|} \sum_{i \in N_{\gamma_j'}} \mathbb{E}\left\|\hat{\boldsymbol{g}}_t^{[\gamma_{j-1}':\gamma_j'](i)}\right\|_2^2 + 2\frac{1}{\left|N_{\gamma_j'}\right| K} \sum_{i \in N_{\gamma_j'}} \sum_{k=0}^{K-1} \mathbb{E}\left\|\tilde{\boldsymbol{g}}_{t,k}^{[\gamma_{j-1}':\gamma_j'](i)}\right\|_2^2$$
$$(53)$$

With the inequality above, the LHS of Equation (48) is bounded by

$$\sum_{j=0}^{n} \frac{\binom{N}{A} - \binom{N - \left|N_{\gamma_j'}\right|}{A}}{\binom{N}{A}} \cdot \mathbb{E}\left\|\boldsymbol{v}_t^{[\gamma_{j-1}':\gamma_j']}\right\|_2^2 \leq 2\sum_{j=0}^{n} \frac{\binom{N}{A} - \binom{N - \left|N_{\gamma_j'}\right|}{A}}{\binom{N}{A} \left|N_{\gamma_j'}\right|} \sum_{i \in N_{\gamma_j'}} \mathbb{E}\left\|\hat{\boldsymbol{g}}_t^{[\gamma_{j-1}':\gamma_j'](i)}\right\|_2^2 \quad (54)$$

$$+ 2\sum_{j=0}^{n} \frac{\binom{N}{A} - \binom{N - \left|N_{\gamma_j'}\right|}{A}}{\binom{N}{A} \left|N_{\gamma_j'}\right| K} \sum_{i \in N_{\gamma_j'}} \sum_{k=0}^{K-1} \mathbb{E}\left\|\tilde{\boldsymbol{g}}_{t,k}^{[\gamma_{j-1}':\gamma_j'](i)}\right\|_2^2$$
$$(55)$$

According to Lemma C.2, we have

$$\frac{\binom{N}{A} - \binom{N - \left|N_{\gamma_j'}\right|}{A}}{\binom{N}{A} \left|N_{\gamma_j'}\right|} \leq \frac{A}{N}. \quad (56)$$

Therefore, we further bound Equation (55) for

$$\sum_{j=0}^{n} \frac{\binom{N}{A} - \binom{N - \left|N_{\gamma_j'}\right|}{A}}{\binom{N}{A}} \cdot \mathbb{E}\left\|\boldsymbol{v}_t^{[\gamma_{j-1}':\gamma_j']}\right\|_2^2 \quad (57)$$

$$\leq \frac{2A}{N} \sum_{j=0}^{n} \sum_{i \in N_{\gamma_j'}} \mathbb{E}\left\|\hat{\boldsymbol{g}}_t^{[\gamma_{j-1}':\gamma_j'](i)}\right\|_2^2 + \frac{2A}{NK} \sum_{j=0}^{n} \sum_{i \in N_{\gamma_j'}} \sum_{k=0}^{K-1} \mathbb{E}\left\|\tilde{\boldsymbol{g}}_{t,k}^{[\gamma_{j-1}':\gamma_j'](i)}\right\|_2^2 \quad (58)$$

$$= \frac{2A}{N} \sum_{i \in [N]} \mathbb{E} \left\| \hat{\boldsymbol{g}}_t^{[0:\gamma_i](i)} \right\|_2^2 + \frac{2A}{NK} \sum_{i \in [N]} \sum_{k=0}^{K-1} \mathbb{E} \left\| \tilde{\boldsymbol{g}}_{t,k}^{[0:\gamma_i](i)} \right\|_2^2 \tag{59}$$

where the last equation holds because the second norm is non-zero when the client $i \in [N]$ owns the part $[\gamma'_{j-1} : \gamma'_j], j \in \{0, \ldots, n\}$, and different parts (i.e., $[\gamma'_{j-1} : \gamma'_j]$ for different $j$s) are independent with each other when calculating the second norm.

With Assumption 5.3 and Assumption 5.1, we separately bound two terms in Equation (59) for

$$\left\| \hat{\boldsymbol{g}}_t^{[0:\gamma_i](i)} \right\|_2^2 \le \delta_t^2 \left\| \tilde{\boldsymbol{x}}_t \right\|_2^2; \qquad \left\| \tilde{\boldsymbol{g}}_{t,k}^{[0:\gamma_i](i)} \right\|_2^2 \le L^2 \left\| \boldsymbol{x}_{t,k}^{(i)} - \tilde{\boldsymbol{x}}_t \right\|_2^2. \tag{60}$$

Therefore, With the inequality above, the LHS of Equation (48) is bounded by

$$\sum_{j=0}^n \frac{\binom{N}{A} - \binom{N - \left| N_{\gamma'_j} \right|}{A}}{\binom{N}{A}} \cdot \mathbb{E} \left\| \boldsymbol{v}_t^{[\gamma'_{j-1} : \gamma'_j]} \right\|_2^2 \tag{61}$$

$$\le \frac{2A}{N} \sum_{i \in [N]} \delta_t^2 \left\| \tilde{\boldsymbol{x}}_t \right\|_2^2 + \frac{2A}{NK} \sum_{i \in [N]} \sum_{k=0}^{K-1} L^2 \cdot \mathbb{E} \left\| \boldsymbol{x}_{t,k}^{(i)} - \tilde{\boldsymbol{x}}_t \right\|_2^2 \tag{62}$$

$$\le 2A\delta_t^2 \left\| \tilde{\boldsymbol{x}}_t \right\|_2^2 + 72KA\eta_l^2 L^2 \left( \delta_t^2 \left\| \tilde{\boldsymbol{x}}_t \right\|_2^2 + \sigma^2 + \left\| \nabla F(\tilde{\boldsymbol{x}}_t) \right\|_2^2 \right) \tag{63}$$

Since the learning rate meets the constraint of Lemma C.3, Equation (63) is obtained when Lemma C.3 applies. Furthermore, with the defined learning rate, we can attain the desired conclusion of Equation (48). $\qquad \square$

**Lemma C.5.** *Suppose that Assumption 5.1, 5.2 and 5.3 hold. Let the local learning rate satisfy* $\eta_l \le \min \left( \frac{1}{2L\sqrt{K(K+1)}}, \frac{1}{6L\sqrt{(K+1)A}} \right)$. *With FIARSE, we have the following conclusion:*

$$\mathbb{E} \left\| \tilde{\boldsymbol{x}}_{t+1} - \tilde{\boldsymbol{x}}_t \right\|_2^2 \le 8\eta_s^2 \eta_l^2 K^2 \left( A\delta_t^2 \left\| \tilde{\boldsymbol{x}}_t \right\|_2^2 + \left\| \nabla F(\tilde{\boldsymbol{x}}_t) \right\|_2^2 + \sigma^2 \right) \tag{64}$$

*Proof.* The recursive function of the global updates of FIARSE follows that

$$\tilde{\boldsymbol{x}}_{t+1} - \tilde{\boldsymbol{x}}_t = -\eta_s \eta_l \mathsf{Agg}_{i \in \mathcal{A}} \left( \sum_{k=0}^{K-1} \nabla_{\boldsymbol{x}_{t,k}^{(i)}} F_i \left( \boldsymbol{x}_{t,k}^{(i)} \odot \mathcal{M}_t^{(i)} \left( \boldsymbol{x}_{t,k}^{(i)} \right) \right) \right). \tag{65}$$

Let us define $\boldsymbol{g}_{t,k}^{[\gamma'_{j-1} : \gamma'_j](i)}$ to be

$$\boldsymbol{g}_{t,k}^{[\gamma'_{j-1} : \gamma'_j](i)} = \nabla_{\boldsymbol{x}_{t,k}^{(i)}} F_i^{[\gamma'_{j-1} : \gamma'_j]} \left( \boldsymbol{x}_{t,k}^{(i)} \odot \mathcal{M}_t^{(i)} \left( \boldsymbol{x}_{t,k}^{(i)} \right) \right). \tag{66}$$

Therefore,

$$\mathsf{Agg}_{i \in \mathcal{A}} \left( \sum_{k=0}^{K-1} \nabla_{\boldsymbol{x}_{t,k}^{(i)}} F_i \left( \boldsymbol{x}_{t,k}^{(i)} \odot \mathcal{M}_t^{(i)} \left( \boldsymbol{x}_{t,k}^{(i)} \right) \right) \right) = \bigcup_{j \in [n]} \frac{1}{\left| \mathcal{A}_{\gamma'_j} \right|} \sum_{i \in \mathcal{A}_{\gamma'_j}} \sum_{k=0}^{K-1} \boldsymbol{g}_{t,k}^{[\gamma'_{j-1} : \gamma'_j](i)}. \tag{67}$$

The bound for $\mathbb{E} \left\| \tilde{\boldsymbol{x}}_{t+1} - \tilde{\boldsymbol{x}}_t \right\|_2^2$ is formulated and simplified as follows:

$$\mathbb{E} \left\| \tilde{\boldsymbol{x}}_{t+1} - \tilde{\boldsymbol{x}}_t \right\|_2^2 = \mathbb{E} \sum_{j=0}^n \left\| \frac{\eta_s \eta_l}{\left| \mathcal{A}_{\gamma'_j} \right|} \sum_{i \in \mathcal{A}_{\gamma'_j}} \sum_{k=0}^{K-1} \boldsymbol{g}_{t,k}^{[\gamma'_{j-1} : \gamma'_j](i)} \right\|_2^2 \tag{68}$$

$$\le \eta_s^2 \eta_l^2 \cdot \mathbb{E} \sum_{j=0}^n \frac{K}{\left| \mathcal{A}_{\gamma'_j} \right|} \sum_{i \in \mathcal{A}_{\gamma'_j}} \sum_{k=0}^{K-1} \left\| \boldsymbol{g}_{t,k}^{[\gamma'_{j-1} : \gamma'_j](i)} \right\|_2^2 \tag{69}$$

$$= \eta_s^2 \eta_l^2 K \cdot \sum_{j=0}^{n} \sum_{i \in N_{\gamma_j'}} \sum_{k=0}^{K-1} \frac{1}{\left|N_{\gamma_j'}\right|} \cdot \frac{\binom{N}{A} - \binom{N - \left|N_{\gamma_j'}\right|}{A}}{\binom{N}{A}} \cdot \mathbb{E} \left\| \boldsymbol{g}_{t,k}^{[\gamma_{j-1}':\gamma_j'](i)} \right\|_2^2 \quad (70)$$

Equation (69) is obtained based on Cauchy–Shwarz inequality, and Equation (70) holds according to Lemma C.1.

Let us define

$$\bar{\boldsymbol{g}}_t^{[\gamma_{j-1}':\gamma_j'](i)} = \nabla_{\tilde{\boldsymbol{x}}_t} F_i^{[\gamma_{j-1}':\gamma_j']} \left( \tilde{\boldsymbol{x}}_t \odot \mathcal{M}_t^{(i)} (\tilde{\boldsymbol{x}}_t) \right) - \nabla F_i^{[\gamma_{j-1}':\gamma_j']} (\tilde{\boldsymbol{x}}_t) \odot \mathcal{M}_t^{(i)} (\tilde{\boldsymbol{x}}_t), \quad (71)$$

$$\ddot{\boldsymbol{g}}_t^{[\gamma_{j-1}':\gamma_j'](i)} = \nabla F_i^{[\gamma_{j-1}':\gamma_j']} (\tilde{\boldsymbol{x}}_t) \odot \mathcal{M}_t^{(i)} (\tilde{\boldsymbol{x}}_t) - \nabla F^{[\gamma_{j-1}':\gamma_j']} (\tilde{\boldsymbol{x}}_t). \quad (72)$$

Since

$$\boldsymbol{g}_{t,k}^{[\gamma_{j-1}':\gamma_j'](i)} = \tilde{\boldsymbol{g}}_{t,k}^{[\gamma_{j-1}':\gamma_j'](i)} + \bar{\boldsymbol{g}}_t^{[\gamma_{j-1}':\gamma_j'](i)} + \ddot{\boldsymbol{g}}_t^{[\gamma_{j-1}':\gamma_j'](i)} + \nabla F^{[\gamma_{j-1}':\gamma_j']} (\tilde{\boldsymbol{x}}_t), \quad (73)$$

where $\tilde{\boldsymbol{g}}_{t,k}^{[\gamma_{j-1}':\gamma_j'](i)}$ is defined in Equation (50), we bound $\left\| \boldsymbol{g}_{t,k}^{[\gamma_{j-1}':\gamma_j'](i)} \right\|_2^2$ by splitting it into four terms:

$$\left\| \boldsymbol{g}_{t,k}^{[\gamma_{j-1}':\gamma_j'](i)} \right\|_2^2 \quad (74)$$

$$\leq 4 \left\| \tilde{\boldsymbol{g}}_{t,k}^{[\gamma_{j-1}':\gamma_j'](i)} \right\|_2^2 + 4 \left\| \bar{\boldsymbol{g}}_t^{[\gamma_{j-1}':\gamma_j'](i)} \right\|_2^2 + 4 \left\| \ddot{\boldsymbol{g}}_t^{[\gamma_{j-1}':\gamma_j'](i)} \right\|_2^2 + 4 \left\| \nabla F^{[\gamma_{j-1}':\gamma_j']} (\tilde{\boldsymbol{x}}_t) \right\|_2^2. \quad (75)$$

Therefore, to bound Equation (70), we should analyze the following inequality, i.e.,

$$\sum_{j=0}^{n} \sum_{i \in N_{\gamma_j'}} \sum_{k=0}^{K-1} \frac{1}{\left|N_{\gamma_j'}\right|} \cdot \frac{\binom{N}{A} - \binom{N - \left|N_{\gamma_j'}\right|}{A}}{\binom{N}{A}} \cdot \mathbb{E} \left\| \boldsymbol{g}_{t,k}^{[\gamma_{j-1}':\gamma_j'](i)} \right\|_2^2 \quad (76)$$

$$\leq 4 \sum_{j=0}^{n} \sum_{i \in N_{\gamma_j'}} \sum_{k=0}^{K-1} \frac{1}{\left|N_{\gamma_j'}\right|} \cdot \frac{\binom{N}{A} - \binom{N - \left|N_{\gamma_j'}\right|}{A}}{\binom{N}{A}} \cdot \mathbb{E} \left\| \tilde{\boldsymbol{g}}_{t,k}^{[\gamma_{j-1}':\gamma_j'](i)} \right\|_2^2 \quad (77)$$

$$+ 4K \cdot \sum_{j=0}^{n} \sum_{i \in N_{\gamma_j'}} \frac{1}{\left|N_{\gamma_j'}\right|} \cdot \frac{\binom{N}{A} - \binom{N - \left|N_{\gamma_j'}\right|}{A}}{\binom{N}{A}} \cdot \mathbb{E} \left\| \bar{\boldsymbol{g}}_t^{[\gamma_{j-1}':\gamma_j'](i)} \right\|_2^2 \quad (78)$$

$$+ 4K \cdot \sum_{j=0}^{n} \sum_{i \in N_{\gamma_j'}} \frac{1}{\left|N_{\gamma_j'}\right|} \cdot \frac{\binom{N}{A} - \binom{N - \left|N_{\gamma_j'}\right|}{A}}{\binom{N}{A}} \cdot \mathbb{E} \left\| \ddot{\boldsymbol{g}}_t^{[\gamma_{j-1}':\gamma_j'](i)} \right\|_2^2 \quad (79)$$

$$+ 4K \cdot \sum_{j=0}^{n} \frac{\binom{N}{A} - \binom{N - \left|N_{\gamma_j'}\right|}{A}}{\binom{N}{A}} \cdot \mathbb{E} \left\| \nabla F^{[\gamma_{j-1}':\gamma_j']} (\tilde{\boldsymbol{x}}_t) \right\|_2^2. \quad (80)$$

There are four terms in the above inequality, i.e., Equation (77), Equation (78), Equation (79), and Equation (80). Subsequently, we analyze these four terms one by one.

• For Equation (77), we apply Lemma C.2 and obtain that

$$\sum_{j=0}^{n} \sum_{i \in N_{\gamma_j'}} \sum_{k=0}^{K-1} \frac{1}{\left|N_{\gamma_j'}\right|} \cdot \frac{\binom{N}{A} - \binom{N - \left|N_{\gamma_j'}\right|}{A}}{\binom{N}{A}} \cdot \mathbb{E} \left\| \tilde{\boldsymbol{g}}_{t,k}^{[\gamma_{j-1}':\gamma_j'](i)} \right\|_2^2 \leq \frac{A}{N} \sum_{j=0}^{n} \sum_{i \in N_{\gamma_j'}} \sum_{k=0}^{K-1} \mathbb{E} \left\| \tilde{\boldsymbol{g}}_{t,k}^{[\gamma_{j-1}':\gamma_j'](i)} \right\|_2^2.$$
$$(81)$$

In Lemma C.4, we mention that

$$\sum_{j=0}^{n} \sum_{i \in N_{\gamma_j'}} \sum_{k=0}^{K-1} \mathbb{E} \left\| \tilde{\boldsymbol{g}}_{t,k}^{[\gamma_{j-1}':\gamma_j'](i)} \right\|_2^2 = \sum_{i \in [N]} \sum_{k=0}^{K-1} \mathbb{E} \left\| \tilde{\boldsymbol{g}}_{t,k}^{[0:\gamma_i](i)} \right\|_2^2; \quad \left\| \tilde{\boldsymbol{g}}_{t,k}^{[0:\gamma_i](i)} \right\|_2^2 \leq L^2 \left\| \boldsymbol{x}_{t,k}^{(i)} - \tilde{\boldsymbol{x}}_t \right\|_2^2.$$

$$(82)$$

Therefore, Equation (77) is further bounded by

$$\sum_{j=0}^{n} \sum_{i \in N_{\gamma_j'}} \sum_{k=0}^{K-1} \frac{1}{\left| N_{\gamma_j'} \right|} \cdot \frac{\binom{N}{A} - \binom{N - \left| N_{\gamma_j'} \right|}{A}}{\binom{N}{A}} \cdot \mathbb{E} \left\| \tilde{\boldsymbol{g}}_{t,k}^{[\gamma_{j-1}':\gamma_j'](i)} \right\|_2^2 \leq \frac{AL^2}{N} \sum_{i \in [N]} \sum_{k=0}^{K-1} \mathbb{E} \left\| \boldsymbol{x}_{t,k}^{(i)} - \tilde{\boldsymbol{x}}_t \right\|_2^2.$$

$$(83)$$

- Similarly, Equation (78) is bounded for

$$\sum_{j=0}^{n} \sum_{i \in N_{\gamma_j'}} \frac{1}{\left| N_{\gamma_j'} \right|} \cdot \frac{\binom{N}{A} - \binom{N - \left| N_{\gamma_j'} \right|}{A}}{\binom{N}{A}} \cdot \mathbb{E} \left\| \bar{\boldsymbol{g}}_t^{[\gamma_{j-1}':\gamma_j'](i)} \right\|_2^2 \leq \frac{A}{N} \sum_{i \in [N]} \mathbb{E} \left\| \bar{\boldsymbol{g}}_t^{[0:\gamma_i](i)} \right\|_2^2. \quad (84)$$

In view that the non-zero part of $\nabla F_i^{[0:\gamma_i]}(\tilde{\boldsymbol{x}}_t)$ is equivalent to $\mathcal{M}_t^{(i)}(\tilde{\boldsymbol{x}}_t)$, we apply Assumption 5.3 and attain that

$$\left\| \bar{\boldsymbol{g}}_t^{[0:\gamma_i](i)} \right\|_2^2 \leq \delta_t^2 \left\| \tilde{\boldsymbol{x}}_t \right\|_2^2. \quad (85)$$

Therefore, the bound of Equation (78) is

$$\sum_{j=0}^{n} \sum_{i \in N_{\gamma_j'}} \frac{1}{\left| N_{\gamma_j'} \right|} \cdot \frac{\binom{N}{A} - \binom{N - \left| N_{\gamma_j'} \right|}{A}}{\binom{N}{A}} \cdot \mathbb{E} \left\| \bar{\boldsymbol{g}}_t^{[\gamma_{j-1}':\gamma_j'](i)} \right\|_2^2 \leq A\delta_t^2 \left\| \tilde{\boldsymbol{x}}_t \right\|_2^2. \quad (86)$$

- For Equation (79), we apply $\frac{\binom{N}{A} - \binom{N - \left| N_{\gamma_j'} \right|}{A}}{\binom{N}{A}} \leq 1$ and Assumption 5.2 and obtain

$$\sum_{j=0}^{n} \sum_{i \in N_{\gamma_j'}} \frac{1}{\left| N_{\gamma_j'} \right|} \cdot \frac{\binom{N}{A} - \binom{N - \left| N_{\gamma_j'} \right|}{A}}{\binom{N}{A}} \cdot \mathbb{E} \left\| \ddot{\boldsymbol{g}}_t^{[\gamma_{j-1}':\gamma_j'](i)} \right\|_2^2 \leq \sigma^2 \quad (87)$$

- With $\frac{\binom{N}{A} - \binom{N - \left| N_{\gamma_j'} \right|}{A}}{\binom{N}{A}} \leq 1$, Equation (80) is bounded and simplified for

$$\sum_{j=0}^{n} \frac{\binom{N}{A} - \binom{N - \left| N_{\gamma_j'} \right|}{A}}{\binom{N}{A}} \cdot \mathbb{E} \left\| \nabla F^{[\gamma_{j-1}':\gamma_j']}(\tilde{\boldsymbol{x}}_t) \right\|_2^2 \leq \sum_{j=0}^{n} \mathbb{E} \left\| \nabla F^{[\gamma_{j-1}':\gamma_j']}(\tilde{\boldsymbol{x}}_t) \right\|_2^2 = \left\| \nabla F(\tilde{\boldsymbol{x}}_t) \right\|_2^2$$

$$(88)$$

In conclusion, Equation (70) is bounded by

$$\sum_{j=0}^{n} \sum_{i \in N_{\gamma_j'}} \sum_{k=0}^{K-1} \frac{1}{\left| N_{\gamma_j'} \right|} \cdot \frac{\binom{N}{A} - \binom{N - \left| N_{\gamma_j'} \right|}{A}}{\binom{N}{A}} \cdot \mathbb{E} \left\| \boldsymbol{g}_{t,k}^{[\gamma_{j-1}':\gamma_j'](i)} \right\|_2^2 \quad (89)$$

$$\leq \frac{4AL^2}{N} \sum_{i \in [N]} \sum_{k=0}^{K-1} \mathbb{E} \left\| \boldsymbol{x}_{t,k}^{(i)} - \tilde{\boldsymbol{x}}_t \right\|_2^2 + 4KA\delta_t^2 \left\| \tilde{\boldsymbol{x}}_t \right\|_2^2 + 4K\sigma^2 + 4K \left\| \nabla F(\tilde{\boldsymbol{x}}_t) \right\|_2^2 \quad (90)$$

By applying Lemma C.3, we further simplify the inequality for

$$\sum_{j=0}^{n} \sum_{i \in N_{\gamma'_j}} \sum_{k=0}^{K-1} \frac{1}{\left|N_{\gamma'_j}\right|} \cdot \frac{\binom{N}{A} - \binom{N-\left|N_{\gamma'_j}\right|}{A}}{\binom{N}{A}} \cdot \mathbb{E}\left\|\boldsymbol{g}_{t,k}^{[\gamma'_{j-1}:\gamma'_j](i)}\right\|_2^2 \tag{91}$$

$$\leq 4KA\left(1 + 36\eta_l^2 KL\right)\delta_t^2 \|\tilde{\boldsymbol{x}}_t\|_2^2 + 4K\left(1 + 36\eta_l^2 KAL\right)\sigma^2 + 4K\left(1 + 36\eta_l^2 KAL\right)\|\nabla F(\tilde{\boldsymbol{x}}_t)\|_2^2 \tag{92}$$

By applying the above learning rate, we can attain the desired conclusion. $\qquad\square$

### C.3   Main Proof of Theorem 5.4

As we set $F(\tilde{\boldsymbol{x}})$ with the mask $\mathcal{M} = \mathbf{1}^{N \times d}$, Assumption 5.1 is reduced to the statement that for all $v, \bar{v} \in \mathbb{R}^d$,

$$\|\nabla F_i(v) - \nabla F_i(\bar{v})\|_2 \leq L\|v - \bar{v}\|_2, \quad \forall i \in [M].$$

Therefore, the global objective function $F(\cdot)$ is a L-smooth function. As a result, we have

$$\mathbb{E}F(\tilde{\boldsymbol{x}}_{t+1}) - F(\tilde{\boldsymbol{x}}_t) \leq \mathbb{E}\left\langle\nabla F(\tilde{\boldsymbol{x}}_t), \tilde{\boldsymbol{x}}_{t+1} - \tilde{\boldsymbol{x}}_t\right\rangle + \frac{L}{2}\mathbb{E}\|\tilde{\boldsymbol{x}}_{t+1} - \tilde{\boldsymbol{x}}_t\|_2^2 \tag{93}$$

The iteration function in FIARSE follows that:

- **Local updates:**

$$\boldsymbol{x}_{t,k+1}^{(i)} = \boldsymbol{x}_{t,k}^{(i)} - \eta_l \cdot \nabla_{\boldsymbol{x}_{t,k}^{(i)}} F_i\left(\boldsymbol{x}_{t,k}^{(i)} \odot \mathcal{M}_t^{(i)}\left(\boldsymbol{x}_{t,k}^{(i)}\right)\right) \tag{94}$$

- **Global update:**

$$\tilde{\boldsymbol{x}}_{t+1} - \tilde{\boldsymbol{x}}_t = -\eta_s \eta_l \mathsf{Agg}_{i \in \mathcal{A}}\left(\sum_{k=0}^{K-1} \nabla_{\boldsymbol{x}_{t,k}^{(i)}} F_i\left(\boldsymbol{x}_{t,k}^{(i)} \odot \mathcal{M}_t^{(i)}\left(\boldsymbol{x}_{t,k}^{(i)}\right)\right)\right) \tag{95}$$

Similar to Lemma C.5, we define $\boldsymbol{g}_{t,k}^{[\gamma'_{j-1}:\gamma'_j](i)}$ to be

$$\boldsymbol{g}_{t,k}^{[\gamma'_{j-1}:\gamma'_j](i)} = \nabla_{\boldsymbol{x}_{t,k}^{(i)}} F_i^{[\gamma'_{j-1}:\gamma'_j]}\left(\boldsymbol{x}_{t,k}^{(i)} \odot \mathcal{M}_t^{(i)}\left(\boldsymbol{x}_{t,k}^{(i)}\right)\right). \tag{96}$$

Therefore,

$$\mathsf{Agg}_{i \in \mathcal{A}}\left(\sum_{k=0}^{K-1} \nabla_{\boldsymbol{x}_{t,k}^{(i)}} F_i\left(\boldsymbol{x}_{t,k}^{(i)} \odot \mathcal{M}_t^{(i)}\left(\boldsymbol{x}_{t,k}^{(i)}\right)\right)\right) = \bigcup_{j \in [n]} \frac{1}{\left|\mathcal{A}_{\gamma'_j}\right|} \sum_{i \in \mathcal{A}_{\gamma'_j}} \sum_{k=0}^{K-1} \boldsymbol{g}_{t,k}^{[\gamma'_{j-1}:\gamma'_j](i)}. \tag{97}$$

In the proposed algorithm, the parameters are updated only when the submodels hold the counterpart. In this means, the definition for $\nabla F^{[\gamma'_{j-1}:\gamma'_j]}(\tilde{\boldsymbol{x}}_t)$ is

$$\nabla F^{[\gamma'_{j-1}:\gamma'_j]}(\tilde{\boldsymbol{x}}_t) = \frac{1}{\left|N_{\gamma'_j}\right|} \sum_{i \in N_{\gamma'_j}} \nabla F_i^{[\gamma'_{j-1}:\gamma'_j]}(\tilde{\boldsymbol{x}}_t) \tag{98}$$

Therefore, by applying Lemma C.1, we have

$$\mathbb{E}\left\langle\nabla F(\tilde{\boldsymbol{x}}_t), \tilde{\boldsymbol{x}}_{t+1} - \tilde{\boldsymbol{x}}_t\right\rangle = \sum_{j=0}^{n} \mathbb{E}\left\langle\nabla F^{[\gamma'_{j-1}:\gamma'_j]}(\tilde{\boldsymbol{x}}_t), \tilde{\boldsymbol{x}}_{t+1}^{[\gamma'_{j-1}:\gamma'_j]} - \tilde{\boldsymbol{x}}_t^{[\gamma'_{j-1}:\gamma'_j]}\right\rangle \tag{99}$$

$$= \sum_{j=0}^{n} \mathbb{E}\left\langle\nabla F^{[\gamma'_{j-1}:\gamma'_j]}(\tilde{\boldsymbol{x}}_t), -\eta_s \eta_l \cdot \frac{1}{\left|N_{\gamma'_j}\right|} \cdot \frac{\binom{N}{A} - \binom{N-\left|N_{\gamma'_j}\right|}{A}}{\binom{N}{A}} \sum_{i \in N_{\gamma'_j}} \sum_{k=0}^{K-1} \boldsymbol{g}_{t,k}^{[\gamma'_{j-1}:\gamma'_j](i)}\right\rangle \tag{100}$$

$$= -\eta_s \eta_l K \sum_{j=0}^{n} \frac{\binom{N}{A} - \binom{N - \left|N_{\gamma_j'}\right|}{A}}{\binom{N}{A}} \mathbb{E} \left\langle \nabla F^{[\gamma_{j-1}':\gamma_j']}\left(\tilde{\boldsymbol{x}}_t\right), \frac{1}{\left|N_{\gamma_j'}\right| K} \sum_{i \in N_{\gamma_j'}} \sum_{k=0}^{K-1} \boldsymbol{g}_{t,k}^{[\gamma_{j-1}':\gamma_j'](i)} \right\rangle \quad (101)$$

$$\leq -\frac{\eta_s \eta_l K}{2} \sum_{j=0}^{n} \frac{\binom{N}{A} - \binom{N - \left|N_{\gamma_j'}\right|}{A}}{\binom{N}{A}} \left\| \nabla F^{[\gamma_{j-1}':\gamma_j']}\left(\tilde{\boldsymbol{x}}_t\right) \right\|_2^2 \quad (102)$$

$$+ \frac{\eta_s \eta_l K}{2} \sum_{j=0}^{n} \frac{\binom{N}{A} - \binom{N - \left|N_{\gamma_j'}\right|}{A}}{\binom{N}{A}} \cdot \mathbb{E} \left\| \nabla F^{[\gamma_{j-1}':\gamma_j']}\left(\tilde{\boldsymbol{x}}_t\right) - \frac{1}{\left|N_{\gamma_j'}\right| K} \sum_{i \in N_{\gamma_j'}} \sum_{k=0}^{K-1} \boldsymbol{g}_{t,k}^{[\gamma_{j-1}':\gamma_j'](i)} \right\|_2^2 \quad (103)$$

where the last equation is built upon $\langle \boldsymbol{a}, \boldsymbol{b} \rangle = -\frac{1}{2}\|\boldsymbol{a}\|_2^2 - \frac{1}{2}\|\boldsymbol{b}\|_2^2 + \frac{1}{2}\|\boldsymbol{a} - \boldsymbol{b}\|_2^2 \leq -\frac{1}{2}\|\boldsymbol{a}\|_2^2 + \frac{1}{2}\|\boldsymbol{a} - \boldsymbol{b}\|_2^2$. Lemma C.2 mentions that

$$\frac{A}{N} \leq \frac{\binom{N}{A} - \binom{N - \left|N_{\gamma_j'}\right|}{A}}{\binom{N}{A}} \leq 1, \quad (104)$$

and we simplify Equation (103) with the conclusion from Lemma C.4:

$$\mathbb{E} \left\langle \nabla F\left(\tilde{\boldsymbol{x}}_t\right), \tilde{\boldsymbol{x}}_{t+1} - \tilde{\boldsymbol{x}}_t \right\rangle \quad (105)$$

$$\leq -\frac{\eta_s \eta_l K A}{2N} \|\nabla F\left(\tilde{\boldsymbol{x}}_t\right)\|_2^2 + \frac{\eta_s \eta_l K}{2} \left( 4A\delta_t^2 \|\tilde{\boldsymbol{x}}_t\|_2^2 + 72KA\eta_l^2 L^2 \left( \sigma^2 + \|\nabla F\left(\tilde{\boldsymbol{x}}_t\right)\|_2^2 \right) \right) \quad (106)$$

By applying the above local learning rate, we can further simplify the equation for

$$\mathbb{E} \left\langle \nabla F\left(\tilde{\boldsymbol{x}}_t\right), \tilde{\boldsymbol{x}}_{t+1} - \tilde{\boldsymbol{x}}_t \right\rangle \leq -\frac{\eta_s \eta_l K A}{4N} \|\nabla F\left(\tilde{\boldsymbol{x}}_t\right)\|_2^2 + 2\eta_s \eta_l K A \delta_t^2 \|\tilde{\boldsymbol{x}}_t\|_2^2 + 36K^2 A \eta_s \eta_l^3 L^2 \sigma^2 \quad (107)$$

Plugging Equation (107) and Lemma C.5 back to Equation (93), we have

$$\mathbb{E} F(\tilde{\boldsymbol{x}}_{t+1}) - F(\tilde{\boldsymbol{x}}_t) \leq -\frac{\eta_s \eta_l K A}{4N} \|\nabla F\left(\tilde{\boldsymbol{x}}_t\right)\|_2^2 + 2\eta_s \eta_l K A \delta_t^2 \|\tilde{\boldsymbol{x}}_t\|_2^2 + 36K^2 A \eta_s \eta_l^3 L^2 \sigma^2$$
$$+ 4\eta_s^2 \eta_l^2 K^2 L \left( A\delta_t^2 \|\tilde{\boldsymbol{x}}_t\|_2^2 + \|\nabla F\left(\tilde{\boldsymbol{x}}_t\right)\|_2^2 + \sigma^2 \right) \quad (108)$$

With the above learning rate, we reorder the formula of Equation (108) for

$$\mathbb{E} F(\tilde{\boldsymbol{x}}_{t+1}) - F(\tilde{\boldsymbol{x}}_t) \leq -\frac{\eta_s \eta_l K A}{8N} \|\nabla F\left(\tilde{\boldsymbol{x}}_t\right)\|_2^2 + 4\eta_s \eta_l K A \delta_t^2 \|\tilde{\boldsymbol{x}}_t\|_2^2$$
$$+ 4\eta_s \eta_l^2 K \left( \eta_s K L + 9K\eta_l L^2 \right) \sigma^2 \quad (109)$$

By summing Equation (109) for all $t \in \{0, \ldots, T-1\}$, we have:

$$F_* - F(\tilde{\boldsymbol{x}}_0) \leq \mathbb{E} F(\tilde{\boldsymbol{x}}_{T+1}) - F(\tilde{\boldsymbol{x}}_0) = \sum_{t=0}^{T} \left( \mathbb{E} F(\tilde{\boldsymbol{x}}_{t+1}) - F(\tilde{\boldsymbol{x}}_t) \right) \quad (110)$$

$$\leq -\frac{\eta_s \eta_l K A}{8N} \sum_{t=0}^{T-1} \|\nabla F\left(\tilde{\boldsymbol{x}}_t\right)\|_2^2 + 4\eta_s \eta_l K A \sum_{t=0}^{T-1} \delta_t^2 \|\tilde{\boldsymbol{x}}_t\|_2^2 + 4\eta_s \eta_l^2 K \left( \eta_s K L + 9K\eta_l L^2 \right) \sigma^2 T \quad (111)$$

By reorganizing the inequality above, we have:

$$\frac{1}{T} \sum_{t=0}^{T-1} \|\nabla F\left(\tilde{\boldsymbol{x}}_t\right)\|_2^2 \leq \frac{8\left(F(\tilde{\boldsymbol{x}}_0) - F_*\right) N}{\eta_s \eta_l K A T} + \frac{64N}{A} \eta_s \eta_l K L \sigma^2 + \frac{32N}{T} \sum_{t \in [T]} \delta_t^2 \|\tilde{\boldsymbol{x}}_t\|_2^2 \quad (112)$$

Based on Theorem 5.4, when we apply local and global learning rates as described in Corollary 5.5, we can obtain the desired conclusion that the proposed FIARSE can converge to a stationary point at a rate of $O(1/\sqrt{T})$.

Table 3: Hyperparameter Settings

|  | CIFAR-10 | CIFAR-100 | AGNews |
|---|---|---|---|
| Local Epochs | 5 | 5 | 2 |
| Batch Size | 20 | 20 | 20 |
| Communication Rounds | 800 | 800 | 300 |
| Optimizer | SGD | SGD | AdamW |
| Learning rate ($\log_{10}$) | $\{-1, -2\}$ | $\{-1, -2\}$ | $\{-3, -4, -5\}$ |
| Momentum | $\{0.0, 0.9\}$ | $\{0.0, 0.9\}$ | $\{(0.9, 0.95)\}$ |

Table 4: Test accuracy under four different submodel sizes on different datasets, and 20 out of 100 clients participate in the training at each round. To be more specific, the columns from "Local" to "Model (1.0)" evaluate the test accuracy on the local test datasets, while "Global" evaluates the average test accuracy of the global model of four different sizes (1/64, 1/16, 1/4, 1.0) on the global test dataset.

| Method | CIFAR-10 | | | | | | CIFAR-100 | | | | | |
|---|---|---|---|---|---|---|---|---|---|---|---|---|
|  | Local | Model (1/64) | Model (1/16) | Model (1/4) | Model (1.0) | Global | Local | Model (1/64) | Model (1/16) | Model (1/4) | Model (1.0) | Global |
| HeteroFL | 69.93 | 61.40 | 69.02 | 72.36 | 76.76 | 67.92 | 32.23 | 28.32 | 31.52 | 33.96 | 35.12 | 30.30 |
| FedRolex | 68.64 | 53.16 | 67.00 | 71.60 | 82.80 | 66.75 | 33.00 | 21.36 | 34.12 | 36.72 | 39.80 | 31.33 |
| ScaleFL | 72.05 | 68.44 | 71.12 | 70.36 | 78.28 | 67.27 | 39.57 | 37.92 | 39.60 | 41.84 | 38.92 | 37.63 |
| FIARSE | **79.65** | **73.84** | **80.00** | **80.40** | **84.36** | **76.61** | **42.27** | **40.32** | **43.28** | **43.52** | **41.96** | **38.97** |

# D  Additional Experiments

## D.1  More Threshold Selection Strategies

- **Layer-wise threshold** gives a set of thresholds based on the computation cost of each layer $l \in [L]$, where $L$ is the number of layers of the global model. Therefore, the threshold for each layer is $\theta_{i,l} = \mathsf{TopK}_{\gamma_i}(|\tilde{\boldsymbol{x}}_l|)$, and $\theta_i = \{\theta_{i,l}\}_{l \in [L]}$.

- **Sharding-wise threshold** is a way in the middle that partitions the model into several shardings, and each sharding encompasses a couple of consecutive layers. This is designed for the case when the parameters have distinct distributions across the model. In this case, we partition the layers $[L]$ into multiple group, i.e., $\mathcal{L} = \{[l_j : l_{j+1}]\}$. Therefore, the threshold for each sharding is $\theta_{i,[l_j:l_{j+1}]} = \mathsf{TopK}_{\gamma_i}(|\tilde{\boldsymbol{x}}_{[l_j:l_{j+1}]}|)$.

## D.2  Hyper-parameter settings

Table 3 lists the hyperparameters that we use in the experiments. For CV datasets, we adopt vanilla SGD or momentum SGD (with the setting of 0.9) as an optimizer. For the NLP dataset, we fine-tune the pretrained model with AdamW and set the parameters $(\beta_1, \beta_2)$ for $(0.9, 0.95)$. In our experiments, we keep the learning rate constant. There is no weight decay during our training.

## D.3  Experiments Compute Resources

We train the neural network and run the program on a server with 8 NVIDIA A6000 GPUs, an Intel Xeon Gold 6254 CPU, and 256GB RAM. Our codes are running with Python 3.7 and Pytorch 1.8.1.

## D.4  More Experimental Results

**Participation rates of 20%.** Table 4 presents the test accuracy of CIFAR-10 and CIFAR-100 for the case where the participation ratio is 20%. Notably, upon increasing the participation ratio to 20%, FIARSE exhibits even more remarkable performance compared to the default setting, i.e., the participation ratio is 10%, surpassing baselines by at least 7% and 3% for CIFAR-10 and CIFAR-100, respectively.

**System heterogeneity with five different model sizes.** The experiments are conducted with five different model sizes for $\gamma' = \{0.04, 0.16, 0.36, 0.64, 1.0\}$. The allocation of clients to each level is balanced. It's important to note that our proposed method is flexible and can accommodate varying numbers of complexity levels or client distributions.

Table 5: Test accuracy under five different submodel sizes on different datasets. To be more specific, the columns from "Local" to "Model (1.0)" evaluate the test accuracy on the local test datasets, while "Global" evaluates the average test accuracy of the global model of five different sizes (0.04, 0.16, 0.36, 0.64, 1.0) on the global test dataset.

| Method | CIFAR-10 | | | | | | | CIFAR-100 | | | | | | | AGNews | |
| | Local | Model (0.04) | Model (0.16) | Model (0.36) | Model (0.64) | Model (1.0) | Global | Local | Model (0.04) | Model (0.16) | Model (0.36) | Model (0.64) | Model (1.0) | Global | Local | Global |
|---|---|---|---|---|---|---|---|---|---|---|---|---|---|---|---|---|
| HeteroFL | 72.93 | 64.05 | 71.80 | 75.75 | 77.25 | 75.80 | 70.38 | 35.01 | 30.15 | 33.60 | 36.10 | 37.75 | 37.45 | 32.60 | 90.10 | 89.53 |
| FedRolex | 73.36 | 60.55 | 70.09 | 74.45 | 80.25 | 81.45 | 71.07 | 38.51 | 28.40 | 37.75 | 41.10 | 42.10 | 43.19 | 36.09 | 89.27 | 88.94 |
| ScaleFL | 74.32 | 71.95 | 73.24 | 73.90 | 74.00 | 78.50 | 68.94 | 42.43 | 42.90 | 44.75 | 42.49 | 42.05 | 39.95 | 40.21 | 89.67 | 89.51 |
| FjORD | 74.04 | 73.00 | 73.20 | 73.55 | 72.30 | 78.15 | 72.64 | 43.11 | 42.19 | 43.90 | 45.00 | 42.25 | 42.20 | 40.73 | 90.68 | 89.08 |
| **FIARSE** | **81.79** | **77.75** | **82.15** | **81.40** | **82.90** | **84.75** | **78.13** | **45.94** | **44.35** | **45.65** | **47.80** | **47.00** | **44.90** | **42.61** | **91.55** | **91.50** |

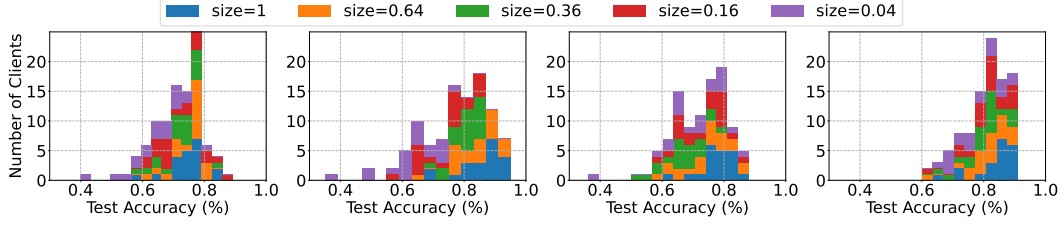

(a) HeteroFL Histogram  (b) FedRolex Histogram  (c) ScaleFL Histogram  (d) FIARSE Histogram

Figure 5: Histograms of various submodel extraction methods on CIFAR-10 under five submodel sizes. Each histogram shows the number of clients achieving different levels of test accuracy.

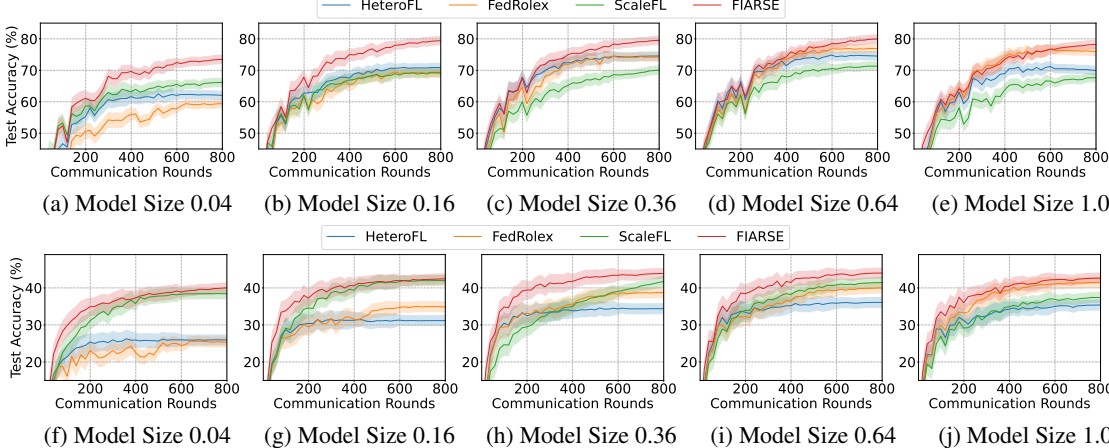

(a) Model Size 0.04  (b) Model Size 0.16  (c) Model Size 0.36  (d) Model Size 0.64  (e) Model Size 1.0

(f) Model Size 0.04  (g) Model Size 0.16  (h) Model Size 0.36  (i) Model Size 0.64  (j) Model Size 1.0

Figure 6: Comparison of test accuracy across communication rounds for different submodel extraction strategies under five varying model sizes (0.04, 0.16, 0.36, 0.64, 1.0) on global test datasets of CIFAR-10 (upper, a – e) and CIFAR-100 (lower, f – j).

Table 5 provides the results under CV and NLP tasks. The results are consistent with the case where four different model sizes were chosen. The proposed FIARSE outperforms all the baselines under both CIFAR-10 and CIFAR-100 in terms of local performance and global performance. As for AGNews, the proposed method still achieves up to 2% improvement over the existing baselines.

Figure 6 presents the model performance of various model sizes over the communication rounds. Similar to the analysis in Section 6.3, the proposed FIARSE significantly outperforms other baselines, especially under a submodel with small sizes. The results demonstrate the superiority of our proposed work in the scenario that we should apply the submodel to the global dataset.

**Ablation study: Effectiveness of TCB-GD.** Table 6 and 7 presents the effectiveness by comparing our proposed work with pruning-greedy [81]. As mentioned before, pruning-greedy is an approach that chooses the largest few values at the beginning of the local training and keeps the mask unchanged during local model training. Apparently, this method does not optimize the model parameters in terms of their importance. A main takeaway in our experimental results is the necessity of optimizing

Table 6: Test accuracy under four different submodel sizes on CIFAR-100 for ablation study. To be more specific, the columns within "Local" evaluate the test accuracy on the local test datasets, while "Global" evaluates the test accuracy of the global model on the global test dataset.

| Method | Local | | | | | Gobal | | | | |
| --- | --- | --- | --- | --- | --- | --- | --- | --- | --- | --- |
| | Model (1/64) | Model (1/16) | Model (1/4) | Model (1.0) | Average | Model (1/64) | Model (1/16) | Model (1/4) | Model (1.0) | Average |
| FIARSE | **39.12** | **43.24** | **43.72** | **40.96** | **41.76** | **35.04** | **39.53** | **41.22** | **38.71** | **38.63** |
| Pruning-greedy | 34.32 | 39.36 | 41.00 | 38.96 | 38.41 | 30.28 | 35.93 | 38.23 | 36.50 | 35.24 |
| FIARSE (layerwise) | 33.44 | 38.24 | 38.64 | 37.00 | 36.83 | 30.54 | 35.94 | 37.12 | 34.46 | 34.52 |

Table 7: Test accuracy under five different submodel sizes on CIFAR-100 for ablation study. To be more specific, the columns within "Local" evaluate the test accuracy on the local test datasets, while "Global" evaluates the test accuracy of the global model on the global test dataset.

| Method | Local | | | | | | Gobal | | | | | |
| --- | --- | --- | --- | --- | --- | --- | --- | --- | --- | --- | --- | --- |
| | Model (0.04) | Model (0.16) | Model (0.36) | Model (0.64) | Model (1.0) | Average | Model (0.04) | Model (0.16) | Model (0.36) | Model (0.64) | Model (1.0) | Average |
| FIARSE | **44.35** | **45.65** | **47.80** | **47.00** | **44.90** | **45.94** | **40.03** | **42.44** | **43.90** | **44.01** | **42.65** | **42.61** |
| Pruning-Greedy | 39.75 | 43.85 | 44.00 | 43.50 | 42.65 | 42.75 | 34.51 | 39.97 | 40.80 | 42.39 | 40.81 | 39.70 |
| FIARSE (layerwise) | 37.30 | 41.75 | 42.75 | 41.85 | 41.10 | 40.95 | 33.63 | 38.43 | 40.29 | 40.99 | 39.10 | 38.49 |

the model with respect to the parameter's importance. In other words, only optimizing the model parameters cannot reflect their importance upon their absolute values.

**Ablation study: Different Threshold Selection Strategies.** Table 6 and 7 compare the proposed work among various threshold selection strategies. For layerwise one, we can extract the model more balanced, where we preserve a fixed ratio of parameters for each layer. The results indicate that a balanced structure performs worse than the one without a balanced guarantee. For these results, we hypothesize that more parameters should be preserved for the first few layers, while the layers close to the output may have massive redundant parameters. Such a conclusion can be verified by comparing ScaleFL [25] with HeteroFL [13]. This is because ScaleFL preserves more model parameters at the beginning of a few layers while discarding the last few layers.

**Ablation study: Submodel Exploration.** Figure 7 and 8 separately include two client heterogeneity settings, i.e., Figure 7 is $\gamma' = \{1/64, 1/16, 1/4, 1.0\}$ with 25 clients each, and Figure 8 is $\gamma' = \{0.04, 0.16, 0.36, 0.64\}$ with 25 clients each. Figure 7a and 8a show two phenomena: (i) all model sizes will gradually slow down their exploration speeds; (ii) even if the largest model size is smaller than the full model size, the number of untrained parameters will eventually go to zero, meaning that none of the parameters are ignored or deactivated. According to Figure 7b and 8b, the extracted submodel for a given size will gradually stabilize, indicating that a suitable submodel architecture has been found. Moreover, a submodel requiring a larger size makes it easier to obtain a stable architecture.

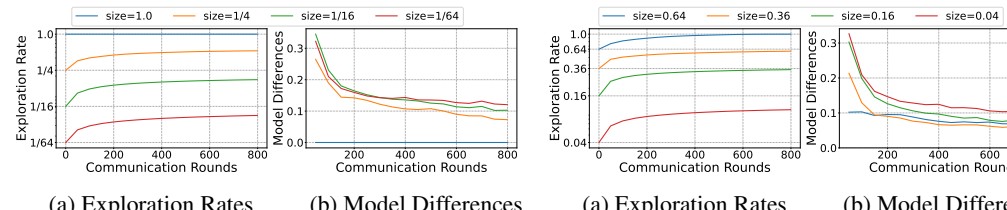

(a) Exploration Rates    (b) Model Differences     (a) Exploration Rates    (b) Model Differences

Figure 7: Exploration rates and model differences against communication rounds for a client heterogeneity setting of $\{1/64, 1/16, 1/4, 1.0\}$.

Figure 8: Exploration rates and model differences against communication rounds for a client heterogeneity setting of $\{0.04, 0.16, 0.36, 0.64\}$.

