# OpenReview forum: "FIARSE: Model-Heterogeneous Federated Learning via Importance-Aware Submodel Extraction"
_NeurIPS.cc/2024/Conference — NeurIPS 2024 poster_

### Official Review · Reviewer_hETm · 2024-07-11

**Soundness:** 3
**Presentation:** 2
**Contribution:** 2
**Rating:** 5
**Confidence:** 3

**Summary:**

This works tackles the known problem of model heterogenity in FL, in which each client is allowed to use a different model according to its computational capabilities, while contributing to learn a global model. The proposed approach (FIARSE) samples submodels out of the global model by estimating the importance of its parameters, and pruning out least important components. The work also provides a theoretical foundation for submodel extraction, even if it is not clear how this analisis relates to the ones proposed in other works. The experiments are carried on CIFAR10/100 (CV task) and AGNews (NLP task).

**Strengths:**

The paper provides a convergence proof for the proposed submodel extraction procedure, matching state-of-art results in term of asymptotic convergence. The experimental setting is appropriate for substantiating the claims of the work, uses datasets and models commonly used in FL (CIFAR10/100 with ResNet-18-GN) and provides results under statistichal heterogeneity.

**Weaknesses:**

1. **FIARSE induces unstructured sparsity in the model.** This is relevant for claims of the paper, because it influences the efficiency of the proposed approach: not all the hardware supports sparse operations, and in general the performance are lower compared to dense models that are actually smaller by the same factor of the induced sparsity. This particularly outstanding given that previous works produce structural sparsity in the given model [1]. As such, this aspect needs to be more clearly stated and analyzed in support of the work's claims: for examples, it would be appropriate to discuss in more detail the computational performance of the proposed method with respect to the others, varying the submodel dimension.
2. **Insufficient comparison with SOTAs.** The proposed approach misses comparison with recent approaches [1,2] which have the advantage of producing structured sparsity in the subsampled model. The works are correctly cited in the introduction, which raises some concers about the choice of excluding them from the evaluation. i believe that those FjORD and NeFL needs to be considered in the evaluation, and better described in the related work section (see next point).
3. **Writing can be improved.** The works presents a discussion on approaches for model-heterogeneous FL in sections 1-2, that is essentially in the introduction and in one sections where the research problem is stated. I found that this structure makes the discussion of the related works not cohesive, resulting in not clear relationship with previous works. Ideally, one would like to understard which are the similar works and how the proposed approach differs from them. In its current form, it's difficult to have such clear idea.
Also on the theory, the work does not detail what is the improvement and/or generalization with respect to previous works [3].
4. **Quality of images.** This is not really I take into account for assessing the paper, but please note that the figures are low-quality so they get grainy on the printed paper.

Points (1-2) are the most important for my evaluation, I believe this could be a good work but the empirical evaluation need to be more convincing in showing an advantage over existing approaches, not only in term of model quality (e.g. accuracy) but also in terms of resources.

[1] Horvath et al., FjORD: Fair and Accurate Federated Learning under heterogeneous targets with Ordered Dropout, NeurIPS 2021

[2] Kang et al., NeFL: Nested Federated Learning for Heterogeneous Clients

[3] Shulgin et al., Towards a Better Theoretical Understanding of Independent Subnetwork Training, ICML 2024

**Questions:**

1. In the introduction, model-heterogeneous FL approaches are divided into static and dynamic submodel extraction, by looking at the costruction of submodels during the training. Why FjORD is regarded as static? In FjORD at each optimization step of portion of contiguous weights are sampled for training according to the client's computational capabilities, so this would make it dynamic based on the above distinction.
2. What is the improvement and/or generalization on the theory part with respect to previous works?

**Limitations:**

Yes

---

> ### Author Response · Authors · 2024-08-07
> **Rebuttal by Authors**
>
> **Response to [W1].** Thanks for your question. While we acknowledge the limitations of unstructured pruning in the *Conclusion*, we also highlight significant industry progress in Appendix A, supported by notable evidence [1,2]. For instance, Nvidia GPUs support unstructured pruning through the NVIDIA Ampere Architecture and NVIDIA TensorRT.
>
> The rest of the response demonstrates the advantages of FIARSE in terms of computation and communication overhead. We train ResNet-18 using CIFAR-10, which is partitioned into 100 clients using Dirichlet distribution with a parameter 0.3. In each communication round, we sample 10 clients, and these clients locally train the model for 4 epochs with a batch size of 20.
>
> According to the numerical results presented in Table 1 and Figure 3 in the manuscript, all baselines can achieve a global accuracy of 65% or higher, and various submodels can realize a global accuracy of 60%. Therefore, the table below presents the number of communication rounds, overall computation, and communication costs when **the average global accuracy reaches 65% and all submodels should achieve an accuracy higher than 60%**:
>
> |Method|Rounds|Total computation costs (TFLOPs)|Total communication costs (GB)|
> |-|-|-|-|
> |HeteroFL|560|4089.59|309.86|
> |FedRolex|NA(640)|4673.82|354.13|
> |ScaleFL|460|5144.41|274.57|
> |FIARSE|**380**|**3620.36**|**210.08**|
>
> Notes to the table: (i) we evaluate the result every 20 rounds, so the value of rounds can be divided by 20. (ii) Model 1/64 of FedRolex never achieves an accuracy above 60%. So we present its result of the 640th round, where FedRolex achieves an average global accuracy of 65% for the first time.
>
> The table demonstrates the superiority of the proposed FIARSE: it requires the fewest communication rounds, and the least communication and computation overhead among the selected methods.
>
> **References:**
> [1] How Well Do Sparse ImageNet Models Transfer?, CVPR'22
> [2] Sparse Fine-tuning for Inference Acceleration of Large Language Models, 2023
>
> ***
>
> **Response to [W2].** Thanks for your suggestion. We conduct empirical studies to train a ResNet with CIFAR-10 using 100 clients, while only 10 clients are selected to train the model in each communication round. The tables below make a comparison between NeFL, FjORD, and the proposed FIARSE.
>
> |Method|Model (1/64)|Model (1/16)|Model (1/4)|Model (1.0)|
> |--|--|--|--|--|
> |NeFL|65.88|72.08|76.52|79.44|
> |FjORD|71.36|71.48|70.04|77.40|
> |FIARSE|73.12|77.20|77.24|82.04|
>
> Among these three methods, the proposed FIARSE significantly outperforms another two baselines.
>
> ***
>
> **Response to [W3/Q2].** Thanks for your suggestion and for bringing interesting work. Due to the space limit, we moved our related work to Appendix A and discussed the similarities and differences compared to the previous works. For example, this work achieves model sparsification in federated learning, but different from the previous studies [1,2], we consider each client to possess heterogeneous computation capacities. We understand that paper organization is not a common practice, and we will insert a brief summarization of related work between *Introduction* and *Preliminary*.
>
> IST [3] focuses on model parallelism in federated learning, where each node independently trains a submodel, and the server assembles all submodels into a global network. In contrast, our proposed FIARSE explores submodel extractions for different model sizes, which can be efficiently trained to achieve remarkable performance. These two works address different problems. From a theoretical perspective, IST analyzes the convergence while training with various submodels. Similar to the conclusion drawn from Theorem 2 of IST, the proposed FIARSE converges to a neighborhood of a stationary point. IST confirms that the bias cannot be eliminated, indicating that the model keeps fluctuating within a constant area. In contrast, our theoretical analysis demonstrates the possibility that FIARSE can minimize this neighborhood area to zero.
>
> **References:**
> [1] Sparse Random Networks for Communication-Efficient Federated Learning, ICLR'23
> [2] DisPFL: Towards Communication-Efficient Personalized Federated Learning via Decentralized Sparse Training, ICML'22
> [3] Towards a Better Theoretical Understanding of Independent Subnetwork Training, ICML'24
>
> ***
>
> **Response to [Q1].** In our opinion, a static model never explores other parameters, while a dynamic model investigates new parts of the model. According to FjORD, the model architecture is fixed at the beginning of training, meaning the submodel never actively explores new parts of the model. Therefore, we consider FjORD to be a static strategy. We will clarify this classification in the Introduction.

---

> > ### Comment · Reviewer_hETm · 2024-08-08
> >
> > I thank the authors for the detailed response. I have read and considered all the reviews and relative rebuttals. I find the authors nicely answered to my concerns, and it is now clearer what is the comparison with the state-of-art in terms of performance on a broader set of metrics, and not just accuracy. As such, I raised my score accordingly, please include the std in the revised manuscript when reporting the results.
> >
> > More as a perplexity than a concern for evaluation, I still don't get the answer to my Q1 and (static vs dynamic submodel extraction) if this classification helps in navigating the literature. Reading again the text, it seems that "static" means that clients' extract the same submodel at each round (see lines 31-32 and Figure 1), while "dynamic" can explore different parameters at different round. To this respect, FjORD samples the parameters to use at each forward pass, so it seems it should be dynamic. It would be nice to further clarify it in the revision.

---

> > > ### Author Response · Authors · 2024-08-11
> > >
> > > Thanks for your feedback. We are happy that our rebuttal has addressed your concerns and appreciate your positive rating.
> > >
> > > Regarding your comments on Question 1, we believe that FjORD can be categorized as either a static or dynamic submodel extraction approach. As you know, FjORD employs ordered dropout, allowing clients to randomly train a submodel within their computational limits at each local step. In our paper, static submodel extraction means that the server consistently extracts a fixed part of a full model and transmits it to a client at every **communication round**. It is reasonable to consider FjORD as a *static* submodel extraction approach since a submodel received by a client is consistently extracted from a specific part of a full model at every **communication round**. On the other hand, FjORD can also be viewed as a *dynamic* submodel extraction approach because a client trains a random portion of the received submodel at each **local training round** (a.k.a. local step).
> > >
> > > From our perspective, debating the category of FjORD is minor; the real challenge lies in understanding why the proposed FIARSE significantly outperforms FjORD, as shown in the table of **Response [W2]**. FjORD adopts ordered dropout to generate submodels of various sizes, which prevents it from constructing optimal submodels for different sizes. In contrast, FIARSE captures the importance of parameters, exploring and building various submodels until they achieve optimal performance for each size.

---

> > > > ### Comment · Reviewer_hETm · 2024-08-14
> > > >
> > > > Dear authors, thanks again for additional clarifications.
> > > > Yes, debating the category of FjORD (as any other competitor) is indeed minor, it was a doubt and not a concern. The explanations you provided will enhance the clarity of the paper in classifying the approaches, given that there is particular accent on FIARSE being dynamic.
> > > >
> > > > Thanks again for the discussion.

---

> ### Author Response · Authors · 2024-08-07
> **Rebuttal by Authors (cont.)**
>
> **Response to [W4].** Thanks for your suggestion. We have updated the experimental figure in the supplementary materials of the rebuttal. I have taken into account the suggestions from Reviewer inPJ and made the necessary updates accordingly, i.e., we will replace Figure 3 in the manuscript with Figure 2 in the supplementary material.

---

### Official Review · Reviewer_dCzP · 2024-07-12

**Soundness:** 3
**Presentation:** 3
**Contribution:** 2
**Rating:** 5
**Confidence:** 4

**Summary:**

The authors address the low device capacities in federated learning. The authors notice that existing submodel extraction methods lack awareness of parameter importance when reducing the model, which may limit the model's performance. They propose an importance-aware dynamic submodel extraction method without the need for additional information beyond the model parameters. Both theoretical and empirical results confirm the effectiveness of the proposed method.

**Strengths:**

1) The topic of model heterogeneity in FL is interesting and meaningful.
2) This work improves previous methods by considering parameter importance when extracting submodels, which is widely overlooked by existing methods.
3) Comprehensive theoretical and empirical results confirm the advantage of the proposed method.

**Weaknesses:**

1) The authors directly masks unimportant parameters during model training, which may not be as efficient as structured-pruning methods (e.g., fedrolex, heterofl). It would be better to take a full comparison on capacity issues like model size, peak memory and FLOPs of different methods.

2) Since only a fraction of model parameters will be assigned with large importance, I'm wondering whether the proposed method will degrade to static submodel extraction where fixed parameters may be always considered important and thus lead to limited model representability. It would be better to contain more analysis of the parameters' importance distributions.

3) Fjord  [6] is an important baseline whose results should be empirically compared.

**Questions:**

Please see weakness

**Limitations:**

Please see weakness

---

> ### Author Rebuttal · Authors · 2024-08-07
>
> **Response to [W1].** Thanks for your question. We train ResNet-18 using CIFAR-10, which is partitioned into 100 clients using a Dirichlet distribution with a parameter of 0.3. In each communication round, we sample 10 clients, and these clients locally train the model for four epochs with a batch size of 20.
>
> According to the numerical results in Table 1 and Figure 3 in the manuscript, all baselines can achieve a global accuracy of 65% or higher, and various submodels can realize a global accuracy of 60%. Therefore, the table below presents the number of communication rounds, overall computation, and communication costs when **the average global accuracy reaches 65% and all submodels should achieve an accuracy higher than 60%**:
>
> |Method|Rounds|Total computation costs (TFLOPs)|Total communication costs (GB)|
> |-|-|-|-|
> |HeteroFL|560|4089.59|309.86|
> |FedRolex|NA(640)|4673.82|354.13|
> |ScaleFL|460|5144.41|274.57|
> |FIARSE|**380**|**3620.36**|**210.08**|
>
> Notes to the table: (i) we evaluate the result every 20 rounds, so the value of rounds can be divided by 20. (ii) Model 1/64 of FedRolex never achieves an accuracy above 60%. So we present its result of the 640th round, where FedRolex achieves an average global accuracy of 65% for the first time.
>
> The table demonstrates the superiority of the proposed FIARSE: it requires the fewest communication rounds, and the least communication and computation overhead among the selected methods.
>
> ***
>
> **Response to [W2].** Thanks for your question. It is possible that our proposed method may eventually extract a constant submodel for a given size. Figure 1(b)(d) in supplementary material shows the differences gradually diminish between two models of $(t-50)$-th and $t$-th communication rounds, where $t \in \{50, 100, \dots\}$. However, it does not mean the performance is limited; instead, the proposed FIARSE finds **optimal model architectures** for various model sizes. This is because all model parameters are explored/trained, as shown in Figure 1(a)\(c), demonstrating the number of parameters that different model sizes have explored.
>
> The following tables present the test accuracy of HeteroFL and the proposed FIARSE on both global and local datasets after 1000 rounds of training: (Note: The numerical cells are in the form of "global/local")
>
> - Figure 1(a)(b): {1/64, 1/16, 1/4, 1.0} with 25 clients each
>
> |Method|Model (1/64)|Model (1/16)|Model (1/4)|Model (1.0)|Avg|
> |-|-|-|-|-|-|
> |HeteroFL|59.70/60.24|66.59/69.32|71.84/72.18|66.08/73.76|66.05/68.88|
> |FIARSE|69.57/73.12|74.93/77.20|76.55/77.24|73.94/82.04|73.75/77.40|
>
> - Figure 1\(c)(d): {0.04, 0.16, 0.36, 0.64} with 25 clients each
>
> |Method|Model (0.04)|Model (0.16)|Model (0.36)|Model (0.64)|Avg|
> |-|-|-|-|-|-|
> |HeteroFL|68.57/70.24|75.91/78.00|78.36/79.32|75.87/83.68|74.68/77.81|
> |FIARSE|77.24/78.76|81.83/82.84|81.73/82.04|79.47/85.80|80.07/82.36|
>
> ***
>
> **Response to [W3].** Thanks for your suggestion. We conducted empirical studies to train a ResNet with CIFAR-10 and CIFAR-100 using 100 clients, while other settings are consistent with those described in the response of **[W1]**. The tables below compare the proposed FIARSE and FjORD.
>
> - CIFAR-10:
>
> |Method|Model (1/64)|Model (1/16)|Model (1/4)|Model (1.0)|
> |-|-|-|-|-|
> |FjORD|71.36|71.48|70.04|77.40|
> |FIARSE|73.12|77.20|77.24|82.04|
>
> - CIFAR-100:
>
> |Method|Model (1/64)|Model (1/16)|Model (1/4)|Model (1.0)|
> |-|-|-|-|-|
> |FjORD|38.16|41.20|40.12|39.84|
> |FIARSE|39.12|43.24|43.72|40.96|
>
> In Table 1 of the supplementary material, we compare FjORD and other baselines comprehensively. We also conduct one more client heterogeneity setting (i.e., {0.04, 0.16, 0.36, 0.64, 1.0} with 20 clients each) and present the numerical results in Table 2.

---

> > ### Author Response · Authors · 2024-08-13
> > **We look forward to your acknowledgement**
> >
> > Dear Reviewer dCzP,
> >
> > We appreciate your high-quality and constructive comments on our work. As we approach the end of the author-reviewer discussion period, we kindly request that you review our response. Feel free to let us know if our response addresses your concerns and if you consider updating the rating. We are more than happy to answer any further questions you may have.
> >
> > Best,
> >
> > Authors

---

> > ### Comment · Reviewer_dCzP · 2024-08-13
> >
> > Thank you for the detailed responses. I think it is still a concern that unstructured pruning might weaken the adaptability of this method in practice since clients' heterogeneity of memory capacity can be one of the most important system constraints in cross-device FL. This response has addressed most of my questions, thus I will raise my rating to 5.

---

> > > ### Author Response · Authors · 2024-08-13
> > >
> > > Dear Reviewer dCzP,
> > >
> > > Thank you for your valuable feedback and for highlighting the hardware limitations of unstructured pruning on resource-constrained devices. While this approach appears cost-effective from a conceptual standpoint, we recognize the practical challenges it presents. We will discuss this limitation in a new section titled *Limitations*.
> > >
> > > We are pleased that our response has largely addressed your concerns, and we appreciate your positive rating.
> > >
> > > Best,
> > >
> > > Authors

---

### Official Review · Reviewer_PQwa · 2024-07-12

**Soundness:** 3
**Presentation:** 2
**Contribution:** 3
**Rating:** 6
**Confidence:** 4

**Summary:**

The authors proposed FIARSE, a model-heterogeneous federated learning framework that addresses the limitations of existing static and dynamic sub-model extraction methods. They introduced an importance aware sub-model extraction technique to extract heterogeneous sub-model from a shared global model. This approach has demonstrated significantly better performance compared to other model-heterogeneous federated learning methods, such as FedRolex and HeteroFL.

**Strengths:**

Strengths:

1. It addresses the important issue of model heterogeneity in federated learning, which is crucial for real-world federated learning applications.
2. The proposed approach efficiently utilizes available resources of clients.
3. The authors provide a theoretical convergence analysis of the proposed method.
4. The paper is well-written and easy to follow.

**Weaknesses:**

Weakness:

1. The performance comparison under different data heterogeneity settings is missing.
2. The authors should discuss the communication cost and computation overhead to reach the target accuracy.
3. The performance evaluation on different ratios of client model heterogeneity distribution is missing.

**Questions:**

1. Can the proposed technique generalize to various data heterogeneity scenarios?
2. Is equal weighting used when aggregating local updates?
3. What will be the impact of client model heterogeneity distribution on global model accuracy??

**Limitations:**

Refer to the weakness.

---

> ### Author Rebuttal · Authors · 2024-08-07
>
> **Response to [W1/Q1].** Thanks for your suggestion. We train ResNet-18 for 800 rounds using CIFAR-10, which is partitioned into 100 clients in a pathological scenario, where each client holds two classes. All these clients are equally distributed into four computation capacities {1/64, 1/16, 1/4, 1.0}. In each communication round, we sample 10 clients, and these clients locally train the model for four epochs with a batch size of 20. The following table presents the test accuracy of HeteroFL and the proposed FIARSE on both global and local datasets: (Note: The numerical cells are in the form of "global/local")
>
> |Method|Model (1/64)|Model (1/16)|Model (1/4)|Model (1.0)|Average|
> |-|-|-|-|-|-|
> |HeteroFL|73.96/75.24|79.04/79.64|81.85/82.24|80.83/81.96|78.92/79.77|
> |FIARSE|81.16/84.16|84.21/85.24|84.95/86.04|84.05/85.20|83.59/85.16|
>
> ***
>
> **Response to [W2]** Thanks for your suggestion. Unlike the experimental setting described in the **Response to [W1/Q1]**, we partition CIFAR-10 into 100 clients using a Dirichlet distribution with a parameter of 0.3.
>
> According to the numerical results presented in Table 1 and Figure 3 in the manuscript, all baselines can achieve a global accuracy of 65% or higher, and various submodels can realize a global accuracy of 60%. Therefore, the table below presents the number of communication rounds, overall computation, and communication costs when **the average global accuracy reaches 65% and all submodels should achieve an accuracy higher than 60%**:
>
> |Method|Rounds|Total computation costs (TFLOPs)|Total communication costs (GB)|
> |-|-|-|-|
> |HeteroFL|560|4089.59|309.86|
> |FedRolex|NA(640)|4673.82|354.13|
> |ScaleFL|460|5144.41|274.57|
> |FIARSE|**380**|**3620.36**|**210.08**|
>
> Note to the table: (i) we evaluate the result every 20 rounds, so the value of rounds can be divided by 20. (ii) Model 1/64 of FedRolex never achieves an accuracy above 60%. So we present its result of the 640th round, where FedRolex achieves an average global accuracy of 65% for the first time.
>
> ***
>
> **Response to [W3/Q3].** Thanks for your suggestion. Different from the experimental setting described in the **response to [W1]**, we partition CIFAR-10 into 100 clients using Dirichlet distribution with a parameter 0.3. Moreover, all these clients are divided into four computation capacities {1/64, 1/16, 1/4, 1.0} with the size of {40, 30, 20, 10}, respectively. The following table presents the test accuracy of HeteroFL and the proposed FIARSE on both global and local datasets: (Note: The numerical cells are in the form of "global/local")
>
> |Method|Model (1/64)|Model (1/16)|Model (1/4)|Model (1.0)|Avg|
> |-|-|-|-|-|-|
> |HeteroFL|67.50/69.70|70.81/70.17|68.83/70.85|60.70/69.00|66.96/70.00|
> |FIARSE|69.96/72.43|73.42/74.63|69.29/74.15|63.17/74.2|68.96/73.61|
>
> ***
>
> **Response to [Q2].** Thanks for your question. The aggregation of a specific parameter depends on the number of clients training in that parameter. For example, if five clients train a parameter, it is updated with their average value. In this sense, equal weighting is used when aggregating local updates. Our method is applicable to the case where clients carry on different weights, and the aggregation rules can follow Pruning-Greedy [1].
>
> **Reference:**
> [1] Every Parameter Matters: Ensuring the Convergence of Federated Learning with Dynamic Heterogeneous Models Reduction, NeurIPS'23

---

> > ### Comment · Reviewer_PQwa · 2024-08-12
> > **Thanks for your response**
> >
> > I thank the authors for providing detailed responses and additional experiments. My questions have been addressed. After reviewing all the feedback and responses to the comments, I'll keep my original rating. Thank you.

---

> > > ### Author Response · Authors · 2024-08-13
> > >
> > > Dear Reviewer PQwa,
> > >
> > > Thank you for your valuable feedback. We are pleased that our response and the additional experiments have addressed your concerns, and we appreciate your positive rating.
> > >
> > > Best,
> > >
> > > Authors

---

### Official Review · Reviewer_inPJ · 2024-07-18

**Soundness:** 3
**Presentation:** 3
**Contribution:** 2
**Rating:** 5
**Confidence:** 4

**Summary:**

This paper tackles the problem of model heterogeneity in federated learning, where clients have different computational capabilities. The authors propose FIARSE, a method that extracts submodels of varying sizes from a global model based on the importance of parameters. The key idea is using a threshold-controlled biased gradient descent to jointly optimize model parameters and determine their importance. They provide theoretical convergence analysis and empirical results showing FIARSE outperforms existing methods.

**Strengths:**

- Extensive experiments demonstrate consistent improvements over baselines across datasets and model architectures.
- The theoretical analysis is solid.
- The method is flexible and can accommodate different numbers of model sizes and client distributions.

**Weaknesses:**

- Because importance is determined by the size of model parameters, smaller parameters may never be selected, which is equivalent to them being 'deactivated.' This seems to have some impact on model performance.
- Similarly, the paper doesn't explore how FIARSE might impact model fairness across different client groups. Could the importance-based extraction inadvertently amplify biases present in the data?
- How sensitive is FIARSE to the choice of threshold? Some analysis on the impact of different thresholding strategies could be valuable.
- Figure 3 seems to only show one set of experimental results. Due to the curves being quite close, they may be influenced by random factors. Can we conduct multiple experiments and display the results as mean + std?

**Questions:**

see Weaknesses

**Limitations:**

yes

---

> ### Author Rebuttal · Authors · 2024-08-07
>
> **Response to [W1].** Thanks for your question. Given an arbitrary neural network, it is impossible to state the optimal architectures for different sizes so that all these submodels can achieve their best performance after federated learning. Therefore, we should **explore different submodel combinations** to find out the best one where all submodels can perform as outstanding as possible. The proposed FIARSE achieves the exploration by means of importance-aware extraction, where we assume the magnitude of model parameters reflects their importance and extract a submodel for a given size with the largest parameters.
>
> In our opinion, your concern focuses on how many parameters are explored by a small model. To elaborate on this, we conduct more experiments to demonstrate the number of parameters that different model sizes have explored at $t$-th round and the model differences between two models of $(t-50)$-th and $t$-th communication rounds, where $t \in \{50, 100, \dots\}$. In specific, we train ResNet-18 on CIFAR-10, which is partitioned across 100 clients using Dirichlet distribution with a parameter $\alpha=0.3$, and ten clients are selected for training at each round. In the supplementary material, Figure 1 presents the numerical results, and the following paragraph will be added to Appendix D.4:
>
> **Ablation study: Submodel Exploration.** *Figure 1 includes two client heterogeneity settings, i.e., Figure 1(a)(b) is {1/64, 1/16, 1/4, 1.0} with 25 clients each, and Figure 1\(c)(d) is {0.04, 0.16, 0.36, 0.64} with 25 clients each. Figure 1(a)\(c) shows two phenomena: (i) all model sizes will gradually slow down their exploration speeds; (ii) even if the largest model size is smaller than the full model size, the number of untrained parameters will eventually go to zero, meaning that none of the parameters are ignored or deactivated. According to Figure 1(b)(d), the extracted submodel for a given size will gradually stabilize, indicating that a suitable submodel architecture has been found. Moreover, a submodel requiring a larger size makes it easier to obtain a stable architecture.*
>
> ***
>
> **Response to [W2].** Thank you for raising an interesting aspect regarding fairness. In our work, model size is decided by client computation capacity. Hence, smaller models can be trained with more data, whereas some parameters in larger models may be trained on limited and biased data. Such limited and biased data could result in unfairness for clients with greater computational capacities, as the extract submodels may be unnecessarily large. However, these clients could extract smaller submodels to achieve a trade-off between model performance and fairness, which is an intriguing area for further investigation. We believe that **the importance-based extraction does not necessarily amplify biases** because the clients with large capacities opt not to use large but unfair models. Additionally, designing new local training approaches is another potential solution to address unfairness issues in our work. We will include this discussion in our paper and consider it for future study.
>
> ***
>
> **Response to [W3].** In this work, we introduce three kinds of thresholding strategies, i.e., model-wise, layer-wise, and sharding-wise. We introduce the model-wise strategy in the main body (Line 156 -- 159), while another two are discussed in Appendix D.1 (Line 1019 -- 1026). In the appendix of the manuscript, Tables 6 and 7 show the empirical results under different client heterogeneity settings, and both tables train a ResNet-18 on the CIFAR-100 dataset:
>
> - Four different model sizes with 25 clients each:
>
> |Strategies|Model (1/64)|Model (1/16)|Model (1/4)|Model (1.0)|
> |--|--|--|--|--|
> |Model-wise|35.04%|39.53%|41.22%|38.71%|
> |Layer-wise|30.54%|35.94%|37.12%|34.46%|
>
> - Five different model sizes with 20 clients each:
>
> |Strategies|Model (0.04)|Model (0.16)|Model (0.36)|Model (0.64)|Model (1.0)|
> |--|--|--|--|--|--|
> |Model-wise|40.03%|42.44%|43.90%|44.01%|42.65%|
> |Layer-wise|33.63%|38.43%|40.29%|40.99%|39.10%|
>
> It is noted that layer-wise strategy divides ResNet-18 into four parts in accordance with the predefined convolutional layers, where each convolutional layer includes two residual blocks. As a result, this strategy can also be regarded as a sharding-wise strategy. Based on the numerical results, we witness that model-wise strategy outperforms layer-wise strategy. More detailed analysis is provided in the Appendix D.4 (Line 1060 -- 1067).
>
> ***
>
> **Response to [W4].** Thanks for your suggestions. We updated Figure 3, which is shown as Figure 2 in the supplementary materials. All these experiments were conducted under three different random seeds. We will update our experimental figures accordingly, including those in the appendix.

---

> > ### Author Response · Authors · 2024-08-12
> > **We look forward to your acknowledgement**
> >
> > Dear Reviewer inPJ,
> >
> > We appreciate your high-quality and constructive comments on our work. As we approach the end of the author-reviewer discussion period, we kindly request that you review our response. Feel free to let us know if our response addresses your concerns and if you consider updating the rating. We are more than happy to answer any further questions you may have.
> >
> > Best,
> >
> > Authors

---

> > ### Comment · Reviewer_inPJ · 2024-08-14
> >
> > Thanks for the detailed rebuttal.
> >
> > Some of my concerns have been addressed.
> >
> > I would raise my score.

---

> ### Author Response · Authors · 2024-08-13
> **We look forward to your response as the discussion period ends in one day**
>
> Dear Reviewer inPJ,
>
> Thank you once again for your commitment to reviewing our paper and assisting us in improving our work. We would like to remind you that the discussion window will close in less than 24 hours, and we eagerly await your feedback.
>
> We have provided detailed explanations for each of your concerns. We would greatly appreciate it if you could review our responses and let us know if they fully or partially address your concerns. Any additional comments you may have would also be highly appreciated.
>
> Best,
>
> Authors

---

> ### Author Response · Authors · 2024-08-14
>
> Dear Reviewer inPJ,
>
> Thank you for your feedback. We are pleased that our rebuttal has addressed your concerns and sincerely appreciate your positive rating.
>
> Best,
>
> Authors

---

### Author Rebuttal · Authors · 2024-08-07

**Figure 1:** We conduct the experiments to demonstrate the number of parameters that have been explored by different model sizes at $t$-th round and the model differences between two models of $(t-50)$-th and $t$-th communication rounds, where $t \in \{50, 100, \dots\}$. In specific, we train ResNet-18 on CIFAR-10, which is partitioned across 100 clients using Dirichlet distribution with a parameter $\alpha=0.3$, and 10 clients are selected for training at each round.

**Figure 2:** An updated version of Figure 3 in the submitted manuscript.

**Table 1** and **Table 2:** An updated version of Table 1 and Table 5 in the submitted manuscript, where we add FjORD as one of our baselines.

---

### Decision · Program_Chairs · 2024-09-25

**Decision:**

Accept (poster)

**Comment:**

The paper is well-written and supported by thorough theoretical analysis and strong experimental results. All reviewers have expressed positive feedback and recommend its acceptance.